



# The role of sublimation as a driver of climate signals in the water isotope content of surface snow: Laboratory and field experimental results

Abigail G. Hughes[1], Sonja Wahl[2], Tyler R. Jones[1], Alexandra Zuhr[3,4], Maria Hörhold[5], James W. C. White[1], and Hans Christian Steen-Larsen[2]

[1]Institute of Arctic and Alpine Research, University of Colorado Boulder, Boulder, Colorado, USA
[2]Geophysical Institute, University of Bergen and Bjerknes Centre for Climate Research, Bergen, Norway
[3]Alfred-Wegener-Institut Helmholtz Zentrum für Polar- und Meeresforschung, Research Unit Potsdam, Telegrafenberg A45, 14473 Potsdam, Germany
[4]University of Potsdam, Institute of Geosciences, Karl-Liebknecht-Str. 24-25, 14476 Potsdam-Golm, Germany
[5]Alfred-Wegener-Institut Helmholtz Zentrum für Polar- und Meeresforschung, Research Unit Bremerhaven, 27568 Bremerhaven, Germany

**Correspondence:** Abigail G. Hughes (abigail.hughes@colorado.edu)

**Abstract.** Ice core water isotope records from Greenland and Antarctica are a valuable proxy for paleoclimate reconstruction, yet the processes influencing the climate signal stored in the isotopic composition of the snow are being revisited. Apart from precipitation input, post-depositional processes such as wind-driven redistribution and vapor-snow exchange processes at and below the surface are hypothesized to contribute to the isotope climate signal. Recent field studies have shown that

surface snow isotopes vary between precipitation events and co-vary with vapor isotopes, which demonstrates that vapor-snow exchange is an important driving mechanism. Here we investigate how vapor-snow exchange and sublimation processes influence the isotopic composition of the snowpack. Controlled laboratory experiments under dry air flow show an increase of snow isotopic composition of up to 8 ‰ $\delta^{18}O$ in the uppermost layer, with an attenuated signal down to 3 cm snow depth over the course of 4-6 days. This enrichment is accompanied by a decrease in the second-order parameter d-excess,

indicating kinetic fractionation processes. Using a simple mass-balance and diffusion box model in conjunction with our observed laboratory vapor isotope signals, we are able to reproduce the observed changes in the snow. This confirms that sublimation alone can lead to a strong enrichment of stable water isotopes in surface snow and subsequent enrichment in the layers below. To compare laboratory experiments with realistic polar conditions, we completed four 2-3 day field experiments at the East Greenland Ice Core Project site (Northeast Greenland) in summer 2019. High-resolution temporal sampling of both

natural and isolated snow was conducted under clear-sky conditions, and demonstrated that the snow isotopic composition changes on hourly timescales. A change of snow isotope content associated with sublimation is currently not implemented in isotope-enabled climate models and is not taken into account when interpreting ice core isotopic records. However, our results demonstrate that post-depositional processes such as sublimation play a role in creating the climate signal recorded in the water isotopes in surface snow. This suggests that the ice core water isotope signal may effectively integrate across multiple

parameters, and the ice core climate record should be interpreted as such.





## 1 Introduction

Water isotope records in polar ice cores have been used as a proxy to reconstruct local temperature and evaporation source conditions dating back hundreds of thousands of years. The isotope-paleothermometer relationship used to interpret ice core water isotope records is based on the assumption that the observed stable water isotope signal is primarily composed of the

input signals from individual precipitation events (Johnsen et al., 2001; Werner et al., 2011; Sime et al., 2019). However, this approach does not take into account the effects of post-depositional surface exchange processes such as vapor exchange and wind-driven redistribution. Recent field studies have shown that the isotopic composition of surface snow varies in parallel with atmospheric water vapor without occurrence of new precipitated snow (Steen-Larsen et al., 2014) and can change on sub-diurnal timescales (Ritter et al., 2016; Casado et al., 2018), suggesting a coupling between the atmospheric water vapor

and surface snow through isotope exchange.

The primary water isotope signal (i.e. $\delta^{18}$O, $\delta$D) in polar precipitation closely reflects the temperature gradient experienced by an air mass from source to deposition, and ultimately the temperature of condensation in the cloud (Dansgaard, 1964; Dansgaard et al., 1973; Jouzel and Merlivat, 1984; Jouzel et al., 1997). Therefore, seasonal differences in the isotopic composition of the precipitation have historically been assumed to be the primary contributor to observed annual cycles in the ice core.

In addition, the second order parameter deuterium excess (d-excess $= \delta$D$-8\cdot\delta^{18}$O) results from kinetic fractionation due to molecular differences between the movement of oxygen and hydrogen in the hydrologic cycle. Traditionally, it is thought that the ice core d-excess signal is driven by the evaporation conditions at the moisture source to an ice core site (Merlivat and Jouzel, 1979).

There are several processes known to influence the climate signal recorded in ice core water isotopes. First, precipitation

may not take place continuously throughout the year, and precipitation intermittency and seasonal bias influence the isotope record (Werner et al., 2000; Casado et al., 2018; Münch et al., 2017; Münch and Laepple, 2018; Zuhr et al., 2021). Second, surface processes such as snowdrift erosion and redistribution may hamper the consecutive deposition and burial of snow layers through time, leading to a lack of a continuous time record. For example, wind-drifted snow can form large persistent surface features/dunes with variations in the snow density and height, altering the isotope signal spatially and vertically. This issue has

been approached by stacking multiple cores or snow pit profiles in order to resolve the climate signal (Casado et al., 2018; Münch et al., 2017; Münch and Laepple, 2018; Zuhr et al., 2021). Third, after deposition the snow and firn undergo vapor diffusion. Unconsolidated snow grains have open pathways between pore spaces, allowing for vapor transport and mixing. Diffusion attenuates the seasonal signal and acts as a smoothing function, and is well-constrained (Whillans and Grootes, 1985; Cuffey and Steig, 1998; Johnsen et al., 2000; Jones et al., 2017). It has been shown that diffusion may smooth across

noise and gaps from intermittent precipitation events, leading to the observed isotope records that imply continuous seasonal temperature changes (Laepple et al., 2016, 2018). However, a remaining missing link between the accumulated signal and the ice core record is a well-defined understanding of snow-air exchange. Continuous isotope exchange between the snow surface and water vapor is known to influence the recorded climate signal, yet the effects are still not fully resolved.





While it was previously assumed that sublimation of snow and ice occurs layer-by-layer and does not cause isotopic frac-
tionation of remaining ice (Dansgaard, 1964), recent studies have shown this is not the case and that snow is subjected to
isotopic fractionation due to sublimation (Ekaykin et al., 2009; Sokratov and Golubev, 2009; Ebner et al., 2017; Madsen et al.,
2019; Beria et al., 2018). In the accumulation zone of ice sheets, the typical region for ice core drill sites, the snow surface and
lower atmosphere are coupled through the continuous humidity exchange in the form of sublimation and deposition of water
molecules and isotopologues (Wahl et al., In review). This interaction continuously imprints on the snow surface $\delta^{18}$O and $\delta$D
isotopic composition and suggests an interpretation of snow isotopes as an integrated climate record, rather than a precipitation
signal only (Steen-Larsen et al., 2014). As sublimation is a non-equilibrium process comparable to evaporation, it likewise
influences the surface snow d-excess, questioning the interpretation of d-excess as a source region signal.

In this study, we investigated how the isotope signal of surface snow is altered over multiple days via post-depositional
exchange between the snow and the near-surface atmospheric water vapor. We utilized three types of experiments from con-
trolled laboratory experiments to in-situ field observations. First, we used a simple laboratory experiment to observe the effects
of sublimation under dry air in a controlled environment. Next, we used two field experiments in northeast Greenland to 1)
analyze the change of snow of known isotopic composition under characteristic polar conditions; and 2) document the isotope
signal evolution of undisturbed snow as it naturally exists at the ice sheet surface. For all experiments, continuous atmospheric
vapor measurements were made above the snow surface to complement the snow sampling and allow us to observe ongoing
snow-vapor isotope exchange. Finally, we reproduced our laboratory observations using a simple box model and parameter-
izations adapted from evaporation processes, supporting the theory that isotope exchange processes are superimposed on the
precipitation input signal. With this data we demonstrate the importance of post-depositional processes on the snow surface
water isotope signal, and provide better constraints on transfer functions between the atmospheric conditions including water
vapor isotopic composition and the climate signal recorded in the surface snow, with implications for interpretation of ice core
records.

## 2   Methods

We investigated through a combination of laboratory and field experiments (Table 1) the influence of phase changes (i.e.
sublimation and vapor deposition) on the snow surface isotopic composition. Laboratory experiments were run in a controlled
environment, which allowed us to isolate the effects of idealized sublimation conditions. The sublimation rate was varied
between different experiment runs through adjusting temperature and the flow rate of dry air. Complementary field experiments
provided greater insight as to how laboratory findings are consistent with field observations occurring at the surface of the ice
sheet. The field experiments were run under close-to-ideal conditions, which limited the duration of the experiments to intervals
of time with clear-sky conditions. In return, the sampling resolution was increased for the field experiments.

Water isotope measurements are reported in standard delta notation given in per mil (‰) (Craig, 1961):

$$\delta_i = \left( \frac{R_i}{R_{VSMOW}} - 1 \right) \times 1000 \qquad (1)$$





**Table 1.** Overview of all experiments conducted. Eight Laboratory experiments (L1-L8) were completed, and four Field experiments (F1-F4). Field experiments included associated Field Box samples (FB2-FB4) and Field Surface samples (FS1-FS4). For L experiments, the controlled settings of the individual experiment runs are given whereas for the field experiments, the environmental conditions are listed. The mean sublimation rate for field (F - FB/FS) experiments was calculated for all observations in which the latent heat flux (LHF) was positive (i.e. directed away from the surface). The median peak sublimation rate in June and July 2019 was 250 $g \cdot m^{-2} day^{-1}$, and maximum observed peak sublimation rates were 600-700 $g \cdot m^{-2} day^{-1}$. Stars denote the subset of laboratory experiments selected for further discussion.

| Experiment type | Experiment name | Mean flow rate (liters per minute) | Mean temperature (°C) | Mean sublimation rate (LHF>0) (g·m$^{-2}$day$^{-1}$) | Peak sublimation rate (LHF>0) (g·m$^{-2}$day$^{-1}$) | Starting snow $\delta^{18}$O (‰) | Duration |
|---|---|---|---|---|---|---|---|
| Laboratory | L1* | 2 | -12 | 428.4 | / | -20 | 5 days |
| | L2 | 3 | -12 | 428.4 | / | -20 | 5 days |
| | L3 | 3 | -12 | 508.4 | / | -20 | 5 days |
| | L4 | 4 | -12 | 568.8 | / | -20 | 5 days |
| | L5* | 5 | -12 | 692.3 | / | -20 | 5 days |
| | L6* | 5 | -9 | 779.9 | / | -28 | 4 days |
| | L7 | 5 | -8 | 965.0 | / | -28 | 6 days |
| | L8 | 5 | -5 | 1329.3 | / | -28 | 5 days |
| Field Box | F2 - FB2 | / | -7.5 | 157.5 | 386.1 | -26.1 | 41 hours |
| | F3 - FB3 | / | -10.5 | 218.5 | 712.9 | -25.2 | 39 hours |
| | F4 - FB4 | / | -12.7 | 130.3 | 279.7 | -25.1 | 51 hours |
| Field Surface | F1 - FS1 | / | -7.6 | 171.6 | 619.6 | / | 57 hours |
| | F2 - FS2 | / | -7.5 | 157.5 | 386.1 | / | 41 hours |
| | F3 - FS3 | / | -10.5 | 218.5 | 712.9 | / | 39 hours |
| | F4 - FS4 | / | -12.7 | 130.3 | 279.7 | / | 51 hours |

where $i$ refers to $\delta$D or $\delta^{18}$O, and $R$ is the ratio of heavy to light isotopes, such that $R_{18O} = [^1H_2^{18}O]/[^1H_2^{16}O]$ and $R_D = [^2H_2^{16}O]/[^1H_2^{16}O]$. Samples were referenced to Vienna Standard Mean Ocean Water (VSMOW).

## 2.1 Laboratory experimental methods (L experiments)

For the laboratory experiments (L1-L8), dry air was circulated over boxes containing isotopically homogeneous snow samples that were kept at fixed temperatures (Fig. 1a). An experimental chamber was designed that consisted of an inner Plexiglass box, which sat inside an outer plywood box used for temperature regulation. The entire setup was placed in a large freezer, with the inner temperature moderated by a PID-controlled heater. Dry air was produced with a generator and run through Drierite desiccant, resulting in humidity <100 ppm. The amount of dry air circulated in the box was regulated by a mass flow controller,





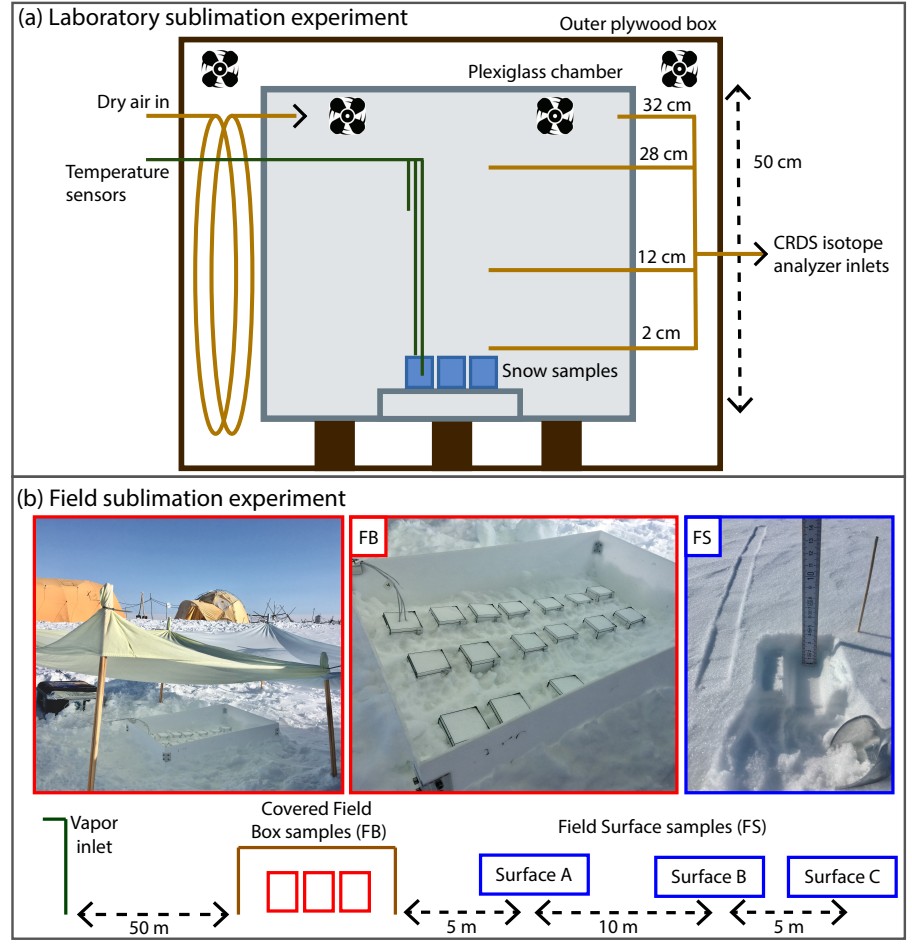

**Figure 1.** (a) Diagram of laboratory experiment setup. A plexiglass chamber was placed within an outer plywood box in a freezer, and dry air was pumped into the inner box above 4-6 homogeneous snow samples, placed on a small shelf to allow airflow. Fans inside the box maintained air circulation. Three temperature sensors were placed at different heights, and continuous CRDS measurements of the vapor were made at four inlet heights (2, 12, 28, 32 cm above the snow surface). (b) Schematic diagram of the field sampling setup at EastGRIP. From left to right: Atmospheric vapor at 10 cm above the snow surface was continuously measured by a CRDS. Homogenous box samples (FB; red) were partially buried and covered, and temperature sensors were placed in the atmosphere, snow surface, and below the surface. Three surface sampling locations (FS A, B, C; blue) were spaced 5-10 m apart, with samples taken at every time interval at each location. A photo example of one sample is shown, in which the left-most sample is the 0-0.5 cm sample, while the intervals for 1-2 and 2-4 cm can be seen in the small pit.

and continuously-running fans maintained mixed air in the chamber. In order to maintain positive pressure in the box, flow rates less than 2 liters per minute (LPM) could not be used. Four to six small boxes (5.7x5.7x7.6 cm) were filled with snow that was well-mixed and sifted so that the snow grain size and isotope value was homogenous, and the initial mass of each sample was





measured. The samples were placed at the bottom of the inner box, on a shelf with underlying airflow to prevent a temperature gradient within the samples. Every 24 hours, one sample was removed, and the mass of that sample was measured. The boxes

could be opened on one side, and a metal spatula was used to collect snow samples with 5 mm resolution to obtain a vertical isotope profile. Snow samples were transferred to 20 mL HDPE scintillation vials for storage and kept frozen until analysis, at which time they were melted and immediately transferred to 2 mL vials. Liquid samples were then analyzed using a Picarro L2130-$i$ cavity ring-down spectrometer (CRDS), in conjunction with a CTC Analytics HTC PAL autosampler injection system and Picarro V1102-$i$ vaporization module. Each sample was measured with six injections, and the reported value is based on

the average of the last three injections to remove memory effects. Every analysis run of 40 samples also included three known water isotope standards bracketing the sample isotope values for calibration (e.g. as done in Jones et al. 2017a). The resulting discrete measurements have uncertainties of 0.1 ‰ $\delta^{18}$O and 1 ‰ $\delta$D.

For the duration of all experiments, several additional parameters were monitored. A Picarro L2130-$i$ CRDS was continuously measuring (~1 Hz) vapor (humidity, $\delta^{18}$O , $\delta$D) from four heights above the snow surface (2, 12, 28, 36 cm), cycling

between each level every hour. The second-order parameter d-excess (d-excess= $\delta$D$-8 \cdot \delta^{18}$O) was also calculated from those measurements. Three Pico Technologies PT-104 Data Logger temperature sensors were placed in the box to record continuously; one 10 cm above the snow surface, one on the surface of the snow, and one ~4 cm below the snow surface.

Two sets of experiments were conducted with varying sublimation rates controlled by adjusting temperature and dry air flow rate (Table 1): For five experiment runs, the temperature was held steady at -12 °C while the dry air flow rate was changed

between a constant flow rate of 2, 3, 4, or 5 LPM (L1-L5, respectively). These experiments used snow from Boulder, Colorado with a starting value of approximately -20 ‰ $\delta^{18}$O, and they were carried out at the Institute of Arctic and Alpine Research at the University of Colorado Boulder. Three additional experiment runs had the temperature of the inner box held constant at -9, -8, or -5 °C (L6-L8, respectively), and the flow rate of dry air above the snow samples was held steady at 5 LPM. These experiments used snow from the East Greenland Ice Core Project field site with a starting $\delta^{18}$O value of approximately -28

‰, and were completed at the section for Physics of Ice, Climate and Earth at the University of Copenhagen. In total, eight experiments were completed.

## 2.2 Field experimental methods (F experiments)

Field experiments were conducted at the East Greenland Ice Core Project (EastGRIP) field camp in July 2019. EastGRIP is located at 75.6268 N, 35.9915 W in the accumulation zone of the Greenland Ice Sheet. In July 2019, the meteorological

conditions at the site were characterized by low temperatures (mean -9.0 °C, measured at 2 m above the snow surface) and high relative humidities (mean 91% RH with respect to ice) leading to an average specific humidity of 2.3 g·kg$^{-1}$. Positive temperatures (>0 °C) were very rare and we observed a positive change in snow height of about 2 cm during July.

The goal of in-situ field experiments was to characterize interactions between the snow surface and near-surface atmospheric vapor on short timescales and to monitor the evolution of the isotopic signal in the snow. To do so, we selected four 2-3 day

experimental periods (Field experiments; F1-F4) during which changes in water isotopes in the snow surface and atmospheric vapor were measured simultaneously through three sample types (Fig. 1b): 1) Discrete box samples (Field Box; FB2-FB4); 2)





Discrete surface samples (Field Surface; FS1-FS4); and 3) Continuous vapor measurements at 10 cm above the snow surface. Each experiment was conducted during a period of good weather, such that precipitation or windblown snow would not bias results. This required air temperatures below freezing, sustained wind speeds below 10-12 knots, and no precipitation.

### 2.2.1 Discrete box samples (FB experiments)

At the beginning of each period, 14-16 boxes (5.7x5.7x7.6 cm) were filled with well-mixed surface snow. Sampling boxes were partially buried in the snow surface, and protected from direct overhead sunlight using a cloth covering. Although this deviates from natural conditions in which the snow surface is exposed and not covered, this modification was necessary to prevent solar heating of the sample boxes which would have otherwise led to melt of the snow not otherwise occurring. A Pico Technologies PT-104 Data Logger was used to measure temperature during the experimental period, with four sensors placed in an additional snow-filled box. The logger continuously recorded temperatures of the ambient air, snow surface, 3 cm below surface, and 6 cm below surface.

One box was collected every three hours; each box was equipped with one removable side so that a vertical profile of the snow was accessible. Snow samples were taken at intervals 0-0.5, 0.5-1, 1-1.5, 1.5-2.5, and 2.5-4.5 cm from the surface using a spatula. The snow samples were transferred to a 20 mL HDPE scintillation vial for storage. Discrete samples are referred to as Field Box (FB) samples, with each experiment designated FB2, FB3, and FB4 (Table 1).

### 2.2.2 Discrete surface samples (FS experiments)

In addition to the isolated boxes, every three hours we collected samples from a clean, undisturbed surface snow area at the same time as the boxes were sampled. Because wind effects can lead to variability in snow surface density and isotopic value, surface samples were collected from three locations, designated sites A, B, and C. The distance from A to B was 10 m, and from B to C was 5 m (Fig. A1). At each surface location, samples were collected from 0-0.5, 0-1, 1-2 and 2-4 cm below the surface (Fig. 1b). The snow samples were transferred to a 20 mL HDPE scintillation vial for storage. Discrete surface samples are referred to as Field Surface (FS) samples, with each experiment designated FS1, FS2, FS3, and FS4 (Table 1).

All snow samples (both Field Box (FB) and Field Surface (FS)) were kept frozen after collection, and were measured at the Stable Isotope Lab at the University of Colorado Boulder. Samples were analyzed using a Picarro L2130-$i$, in conjunction with a CTC Analytics HTC PAL autosampler injection system and Picarro V1102-$i$ vaporization module. The same measurement protocol was used as described in Section 2.1.

### 2.2.3 Continuous vapor measurements

Continuous atmospheric water vapor isotope measurements were made at 10 cm above the snow surface, ∼50 m away from the FS sampling site so as not to be contaminated by snow sampling activity. The measurements were made with a Picarro L2130-$i$ CRDS, which was kept in a temperature-controlled tent and measured humidity, $\delta^{18}$O, and $\delta$D. Using a KNF pump, air was pumped through a ∼12 m long heated copper tube to the analyzer (similar to the setup described in Madsen et al. (2019)).





Four types of calibrations were performed on the water vapor isotope measurements of the CRDS, similar to the calibration protocol described in Steen-Larsen et al. (2013): 1) Humidity calibration; 2) Humidity-isotope response calibration; 3)

VSMOW-VSLAP scale calibration; and 4) Drift calibration. All calibrations are applied to water vapor isotope measurements in both laboratory and field experiments. Details of the calibration setup specific to laboratory and field experiments are described in Appendix B.

Latent heat flux (LHF) was also continuously measured during the field campaign using a Campbell Scientific IRGASON eddy-covariance (EC) system. The 2-in-1 EC system measured humidity and 3-dimensional wind at a sampling frequency of

20 Hz in the same sample volume at 2.15 m above the snow surface. Latent heat flux values were computed for 30-min intervals using Campbell Scientific's software EasyFlux$^{TM}$ adjusted for sublimation conditions and accounting for wind rotation and frequency corrections. Latent heat flux is related to sublimation: Latent heat flux = sublimation rate·$\lambda$, where $\lambda$ is the latent heat of sublimation at 0 °C, 2834 kJ·kg$^{-1}$; a positive latent heat flux indicates sublimation, and negative latent heat flux indicates vapor deposition.

## 2.3 Isotope model

To compare our observations with simulations based on theoretical understanding of the processes involved, we used a numerical isotope box model to simulate the isotope response in the upper 4 cm of snow. Our goal was to demonstrate that a theoretical model based on previously established parameterizations can estimate the isotope response we observed due to sublimation. Because the focus is on the response of the snow to sublimation, the model was driven only using the observations of the mix-

ing ratio and vapor isotopes. Since the snow samples in the experimental setup were subjected to sublimation by introduction of dry air, we can assume that the measured vapor isotope signal was a direct result of sublimation, only. Therefore, we used the vapor measurements and a mass balance equation to calculate the associated change in $\delta^{18}$O and d-excess of the top (0-0.5 cm) layer of snow:

$$R_S^{i+1} = \frac{m^i R_S^i - dm R_E^i}{m^i - dm} \qquad (2)$$

For time $i$, where $R_S$ and $R_E$ are the isotope ratios for snow and sublimate, respectively, $m$ is the mass per unit surface area, and $dm$ is the change in mass. The Johnsen et al. (2000) vapor diffusion model was then applied to the full snow column to estimate the propagation of the signal deeper below the surface. In this case, we assume that sublimation only occurs from the top layer, and any isotopic changes below 0.5 cm can be attributed to diffusion (Ebner et al., 2017). We would expect there to be a very steep isotope gradient within the top layer of snow; therefore, to keep model output consistent and comparable

to laboratory measurements, we used the same vertical depth intervals of 0.5 cm. In order to best replicate experimental results, the model was initiated using experimental conditions including temperature, snow density, and starting snow isotopic composition, and was driven by experimental measurements for vapor observations that change with time (i.e. $\delta^{18}$O, $\delta$D, and humidity). A full description of the model, including associated equations and derivations, is found in Appendix C.



This model allows us to estimate the effect of sublimation on the snow surface and determine the depth at which the signal
propagates into the snow column. Because we can apply this model to both the $\delta^{18}$O and $\delta$D signal, we can also estimate the
d-excess signal. In the following discussion we will focus on model results for snow column $\delta^{18}$O and d-excess in comparison
to laboratory experimental results.

## 3   Results

### 3.1   Laboratory experiments

#### 3.1.1   Sublimation rate

The mean sublimation rate for each laboratory experiment is calculated based on mass loss with time and surface area, reported
in Table 1. Sublimation rate does not significantly change with time, and is shown for each experiment in Fig. A8. The mass of
each box was measured at the onset of the experiment and immediately before each sampling. Since we only push dry air into
the chamber, the experiment relates only to sublimation processes. We find that the latent heat flux associated with sublimation
varies from approximately 15 Wm$^{-2}$ (Experiment L1 at 2 LPM and -12 °C) to 44 Wm$^{-2}$ (Experiment L8 at 5 LPM and -3
°C).

Latent heat flux values in laboratory experiments are comparable to the peak daytime sublimation fluxes observed during
the field campaign, albeit the average latent heat flux during the sublimation period in the field was substantially smaller than
observed during the laboratory experiments (Table 1). The mean daytime positive latent heat flux was ~5 Wm$^{-2}$, and the
maximum latent heat flux observed was ~23 Wm$^{-2}$. Therefore, laboratory experiments can be considered representative of
processes occurring during peak daytime conditions in the field.

#### 3.1.2   Snow measurements

Eight laboratory experiments (L1-L8) were completed, with temperature, dry air flow rate, and sublimation rate documented
in Table 1. In all experiments, the surface snow experiences substantial isotopic changes, with $\delta^{18}$O increasing by up to 8 ‰,
and d-excess decreasing by over 25 ‰. $\delta^{18}$O and d-excess signals for all experiments are shown in Fig. A10, with a subset of
experiments shown in Fig. 2. Changes in the isotope signal are observed to propagate several cm into the snow pack, driven
by the induced sublimate-related isotope change at the surface. The rate of change is calculated for the mean isotope value for
each day of sampling, ranging from 0.25-0.70 ‰ day$^{-1}$ for $\delta^{18}$O and 0.66-2.64 ‰ day$^{-1}$ for d-excess (Fig. A11a-b). There is
a strong relationship between mass loss and isotope values, with an average R$^2$ = 0.90 for daily mean box $\delta^{18}$O vs. mass loss
for experiments L1-L8. The relationship between sublimation rate and $\delta^{18}$O rate of change has R$^2$ = 0.13, and sublimation rate
vs. d-excess rate of change has R$^2$ = 0.54 (Fig. A11c).

Because $\delta^{18}$O reflects equilibrium fractionation and d-excess is influenced by kinetic fractionation, a comparison of these
variables provides insight to the extent of fractionation effects occurring during sublimation. The slope of d-excess vs. $\delta^{18}$O is
calculated for samples within each box (Fig. 3a), and the slope with time over the duration of each experiment is shown in Fig.

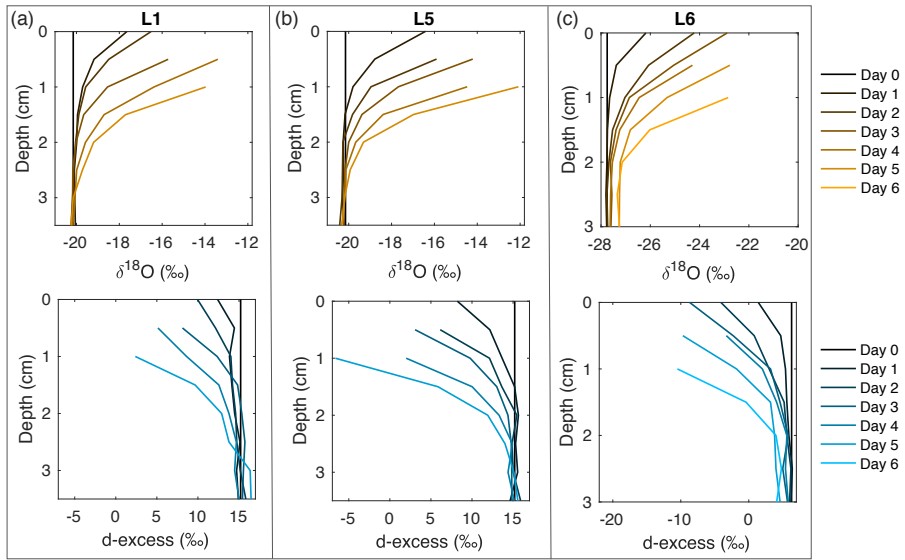

**Figure 2.** Snow $\delta^{18}$O (top) and d-excess (bottom) vertical profiles from three of the laboratory experiments: (a) L1 (-12 °C, 2 LPM); (b) L5 (-12 °C, 5 LPM); and (c) L6 (-9 °C, 5 LPM). Day 0 (black) represents the initial homogeneous snow sample, with colors progressively moving towards orange ($\delta^{18}$O) and blue (d-excess) with each day of sampling. As each experiment progresses from Day 0 to Day 6, sublimation drives an increase in $\delta^{18}$O and decrease in d-excess, with the greatest change at the snow surface. Similar figures for all laboratory experiments (L1-L8) can be found in Fig. A10.

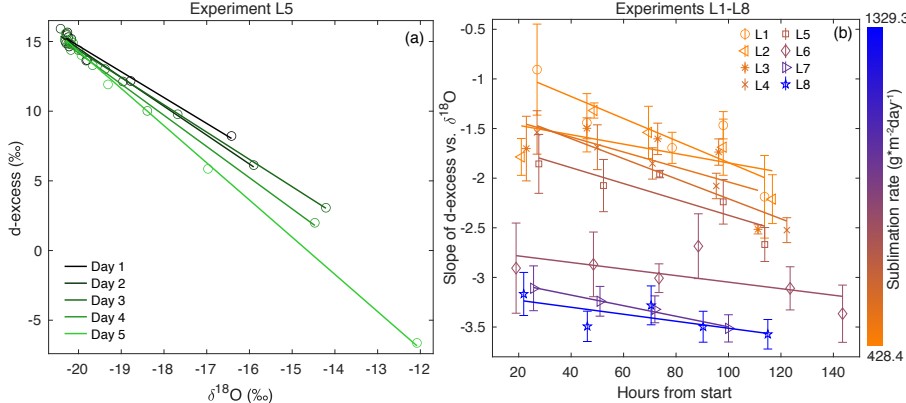

**Figure 3.** (a) d-excess vs. $\delta^{18}$O is shown for the vertical snow profile at each day of sampling in Experiment L5, with a linear regression calculated for each day. This gives a slope of d-excess vs. $\delta^{18}$O, which evolves with time. (b) The slope of d-excess vs. $\delta^{18}$O with time is shown for each experiment L1-L8, demonstrating an inverse relationship between sublimation rate and slope of d-excess vs. $\delta^{18}$O. Error bars indicate 95% confidence intervals for each slope.





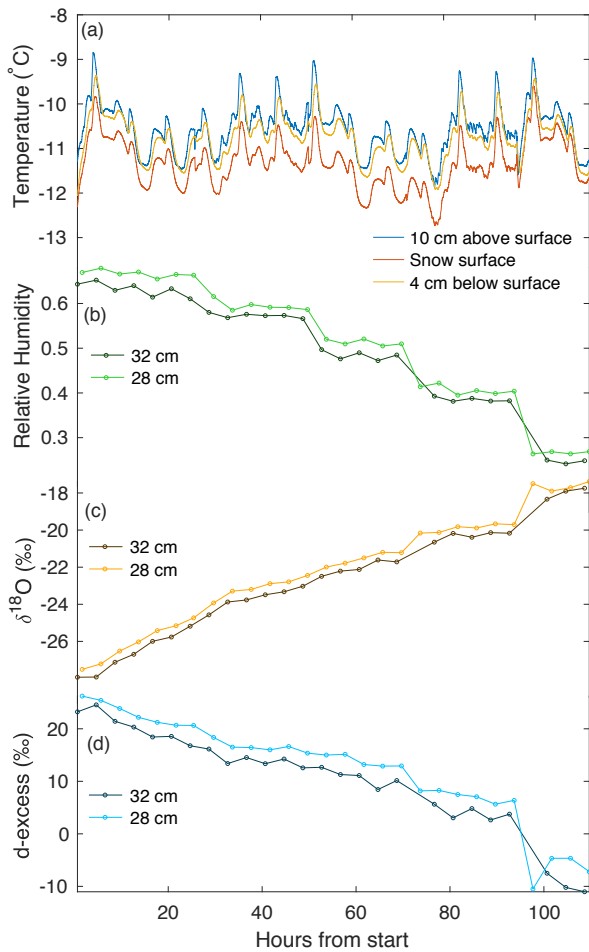

**Figure 4.** An example of continuous temperature and vapor measurements from experiment L5. (a) Three temperature sensors continuously measure at different heights with respect to the snow surface (10 cm above, on the surface, and ∼4 cm below the surface). A CRDS measured (b) humidity; (c) $\delta^{18}$O, and (d) d-excess in vapor, continuously cycling at four heights. (b-d) show vapor measurements at 28 cm and 32 cm above the snow surface. We document the average of each measurement period, with the first 20 minutes excluded to remove memory effects.

3b. The slope ranges from -0.91 to -3.57 ‰ d-excess/‰ $\delta^{18}$O, and decreases over the course of all experiments. In general, there is a decrease in slope associated with an increase in sublimation rate, as indicated by the color scale reflecting sublimation rate in Fig. 3b and as shown in Fig. A9.

### 3.1.3 Vapor measurements

During all laboratory experiments, a Picarro L2130-$i$ CRDS was used to continuously measure vapor in the chamber at 2,
12, 28, and 36 cm, cycling through each height for 1 hour measurement periods. We exclude the first 20 minutes of each





measurement period to remove memory effects from valve changes. Figure 4 shows an example of temperature and vapor data for experiment L5, including the 28 cm and 32 cm levels, which represent sublimated vapor which is more well-mixed than that of immediately above the snow surface. Dry air pumped into the top of the box is mixed using a set of fans creating turbulence above the snow surface. The vertical differences in humidity and isotopic composition of the air in the box (i.e.

differences between 28 cm and 32 cm as seen in Fig. 4) likely indicate that the ventilation is not strong enough to maintain a fully homogeneous air mass in the box, allowing for a slight vertical gradient.

Over the course of each 4-6 day experimental period, we observe several trends in vapor measurements consistent across all laboratory experiments. Humidity decreases with time, due to a reduction in the sublimating surface area each time a snow sample box is removed. Vapor $\delta^{18}O$ increases with time, consistent with the increase in $\delta^{18}O$ observed in the snow surface.

Similarly, d-excess decreases with time in both vapor measurements and the snow surface.

### 3.2 Field experiments

Four experiments (F1, F2, F3, F4) were carried out during the 2019 EastGRIP field season, with surface samples (FS1, FS2, FS3, FS4) collected for all experiments and box samples (FB2, FB3, FB4) collected for three experiments. Each of the four experiments lasted 40-60 hours and is supported by continuous measurements of near-surface (10 cm) atmospheric vapor

($\delta^{18}O$, $\delta D$, d-excess, humidity), temperature (snow and atmosphere), and latent heat flux. Within each experiment, surface snow and box samples are collected every three hours. The duration and average environmental conditions of each experiment is reported in Table 1. A compilation of results for measurements of FS, FB, and atmospheric vapor is shown in Fig. 5 and discussed in the next section. All FS and FB samples are shown in Figs. A12 and A13, respectively, with additional vapor measurements shown in Fig. A14.

#### 3.2.1 Variability in $\delta^{18}O$ and d-excess of surface snow

In order to account for horizontal and vertical spatial variability as a result of redistribution of snow in sastrugi and snow dunes, we averaged isotope values across the three surface locations (A, B, C) for each time of sampling and for each depth interval (i.e. one averaged value each for 0-0.5, 0-1, 1-2, and 2-4 cm at every time sampled). In the following figures and tables we focus on the location-averaged values for each sampling time and depth, referred to as FS1, FS2, FS3, and FS4 for

each FS surface experiment. Isotope values for all surface locations (A, B, C) are shown together with the averages in Fig. A12. Location-averaged surface snow measurements at all depths across FS1, FS2, FS3, and FS4 range from approximately -30.3 to -20.7 ‰ $\delta^{18}O$, and 4.6 to 14.3 ‰ d-excess. In all experiments, we consistently observe changes in surface snow isotopic composition on an hourly timescale (Fig. 5). The maximum change in average $\delta^{18}O$ of the top surface sample (0-0.5 cm) during a single experiment occurred during FS2, which experienced a change of 3 ‰ $\delta^{18}O$ and 4.37 ‰ d-excess. This

evolution is substantially smaller than the isotopic change observed in vapor measurements, which has ranges of 5-12 ‰ $\delta^{18}O$ over individual experiment periods, and ranges for d-excess of 15-24 ‰. The maximum change in $\delta^{18}O$ and d-excess observed within the top surface sample (0.0-5 cm) during each experiment is reported in Table 2.

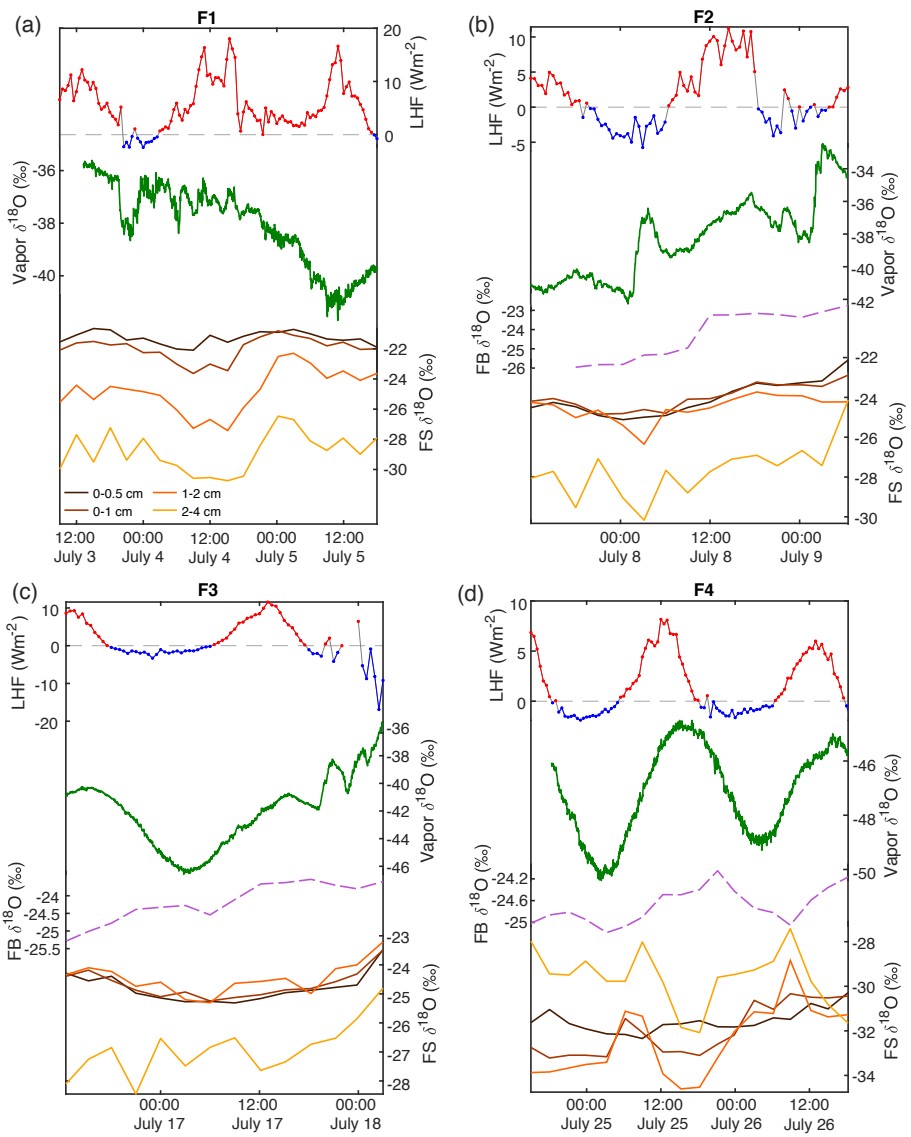

**Figure 5.** A compilation of data from the 2019 field season shows atmospheric measurements and surface snow samples; from top: latent heat flux (red, positive values; blue, negative values; dashed gray line at 0), $\delta^{18}O$ (green) of atmospheric vapor (2 minute average) measured at 10 cm, $\delta^{18}O$ of the top sample (0-0.5 cm) of the FB box sample (pink dashed), and $\delta^{18}O$ of FS snow surface samples. Each snow surface sampling interval shown represents the average of three surface sampling locations (A, B, C) for four different depth intervals: 0-0.5 cm (black), 0-1 cm (red), 1-2 cm (orange), 2-4 cm (yellow). $\delta^{18}O$ of FS samples tends to reflect $\delta^{18}O$ in atmospheric vapor, with the relationship strongest in the upper surface samples (Table 3). Additional data including temperature and vapor humidity are shown in Fig. A14.

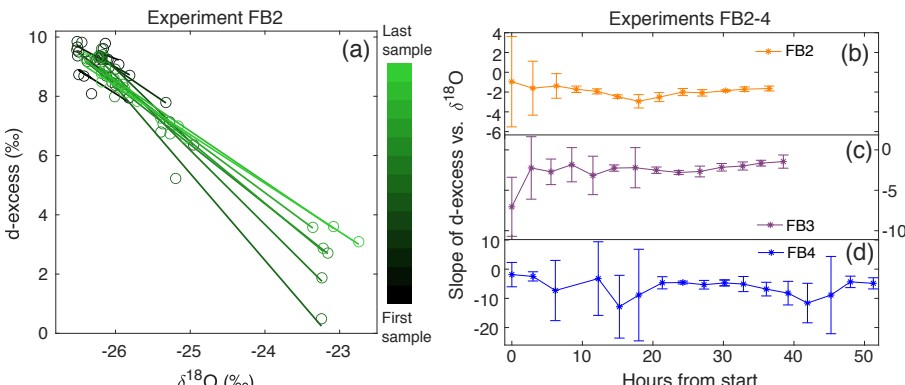

**Figure 6.** (a) d-excess vs. $\delta^{18}O$ is shown for the vertical snow profile at each time of sampling in Experiment FB2, with a linear regression calculated for each day. This gives a slope of d-excess vs. $\delta^{18}O$, which evolves with time. The sampling time is indicated by the color scale from black (first sample taken) to light green (last sample taken). The slope of d-excess vs. $\delta^{18}O$ with time is shown for experiments (b) FB2, (c) FB3, and (d) FB4. Error bars indicate the 95% confidence interval for each slope.

**Table 2.** The maximum range of isotope measurements is shown for the mean value of the top (0-0.5 cm) sample for all FS experiments.

| Experiment | Range $\delta^{18}O$ (‰) | Range d-excess (‰) |
|---|---|---|
| FS1 | 1.44 | 4.00 |
| FS2 | 3.00 | 4.37 |
| FS3 | 1.84 | 4.12 |
| FS4 | 2.06 | 2.62 |

### 3.2.2 Relationship between vapor and surface snow

Over the course of all experiments, the minimum atmospheric vapor $\delta^{18}O$ value observed is -50 ‰, while the maximum observed value is -33 ‰ (a range of 17 ‰). Within each 40-60 hour long experiment, the minimum range of variability observed is about 5 ‰ (F4) and the maximum is about 12 ‰ (F3). Vapor $\delta^{18}O$ co-varies with humidity and temperature (Fig. A14), with lowest $\delta^{18}O$ measurements observed during cold, dry conditions. A clear diurnal cycle is observed in vapor measurements for experiments F3 and F4, while experiments F1 and F2 are more variable. The change in atmospheric $\delta^{18}O$

associated with the diurnal cycle is much smaller than that observed during synoptic weather changes, similar to the pattern previously observed at the Northwest Greenland site NEEM (Steen-Larsen et al., 2014). For example, we observe a strong diurnal cycle in F4 and the first half of F3, both of which have an amplitude of approximately 5-6 ‰ $\delta^{18}O$; the change between experiments with different synoptic-scale atmospheric conditions is much greater (i.e. a 17 ‰ $\delta^{18}O$ range is observed between the maximum value during F2 and the minimum value during F4).

Over clear-sky experimental periods with no precipitation, we observe the $\delta^{18}O$ value of surface snow increasing and decreasing on an hourly timescale, corresponding to changes in vapor $\delta^{18}O$ (Fig. 5). To compare the evolution of the isotope





**Table 3.** The R-value and P-value is documented for the relationship between the top (0-0.5 cm) FS sample $\delta^{18}$O and interpolated vapor $\delta^{18}$O. Significance is determined by p≤0.05. $\delta^{18}$O of vapor vs. surface samples is shown in Fig. A15.

| Experiment | Slope | R-value | P-value | Significance |
|------------|-------|---------|---------|--------------|
| FS1 | 0.84 | 0.22 | 0.374 | No |
| FS2 | 2.16 | 0.73 | 0.002 | Yes |
| FS3 | 4.73 | 0.82 | 0.000 | Yes |
| FS4 | 1.56 | 0.49 | 0.043 | Yes |

signal in vapor and snow measurements, the vapor $\delta^{18}$O is downsampled to 3-hour resolution to match snow sampling resolution. A statistically significant (p ≤ 0.05) relationship is observed between $\delta^{18}$O of 0-0.5 cm snow surface samples and atmospheric vapor measurements for experiments FS2, FS3, and FS4, but not FS1 (Fig. A15, Table 3). The lack of a significant
correlation in FS1 may be a result of some synoptic-scale weather difference, as it is the only experiment period in which there is a sustained decrease in vapor $\delta^{18}$O, and a diurnal cycle in temperature, LHF, and vapor $\delta^{18}$O is least distinguishable.

### 3.3    Isotope model

We provide here simulations using a snow isotope model driven with observations and input from laboratory experiment L5.
The isotope model is run for a 4-day period, with input mimicking experimental conditions. Model output shown in Fig. 7 is run with T = -12 °C, sublimation rate = 700 g·m$^{-2}$day$^{-1}$, and initial snow composition of -20 ‰ $\delta^{18}$O and 15 ‰ d-excess. The evolution of the snow isotope profile is very similar to that observed in laboratory experiments, with increasing $\delta^{18}$O and decreasing d-excess, and downward propagation of the snow isotopic composition influenced by sublimation and subsequent diffusion to a depth of 2-3 cm below the surface.
290       Using the mass balance Eq. 2 paired with the vapor measurements, the model accurately predicts $\delta^{18}$O, but gives spurious results for d-excess. We attribute this difference to uncertainty in the vapor measurements, especially at low humidity levels. As can be seen in the example isotope-humidity response curve (Fig. A5), there is significant instrumental bias and uncertainty at humidity levels below 1000 ppm. At this level, we begin to see larger deviations in the d-excess model output. To account for this, we modify the $\delta$D vapor input by +10 ‰, which produces more realistic changes in the snow isotopic composition.
Using this modification, the model estimates a final (i.e. day 4) top (0-0.5 cm) isotopic composition which has increased by 9.5 ‰ $\delta^{18}$O, and decreased by 16.5 ‰ d-excess (Fig. 7). This adaptation in the model conditions closely matches the final top (0-0.5 cm) composition of experiment L5, which is approximately -12 ‰ $\delta^{18}$O and -5 ‰ d-excess.

### 4    Discussion

In the laboratory experiments, the snow was sublimating under dry air, resulting in a higher latent heat flux than was observed in
a typical field setting. For this reason, laboratory experiments are considered an extreme example of natural processes, and can

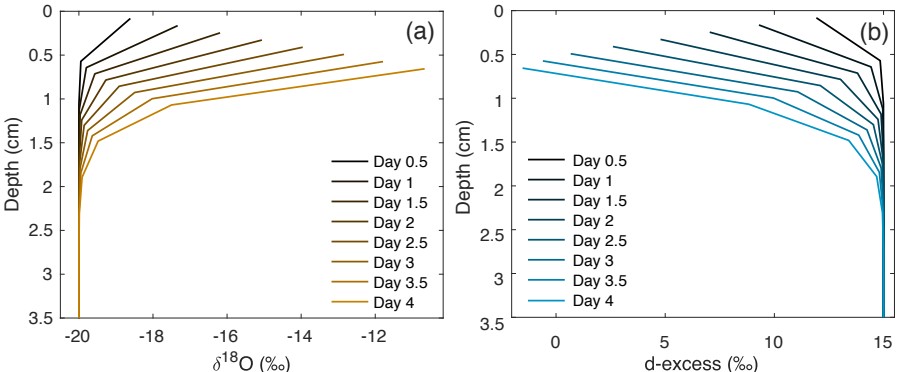

**Figure 7.** Model output of the snow column for (a) $\delta^{18}O$ and (b) d-excess over a 4-day period, with inputs of T = -12 °C, sublimation rate = 700 g·m$^{-2}$day$^{-1}$, and initial snow at homogeneous -20 ‰ $\delta^{18}O$ and 15 ‰ d-excess. Vapor measurements from experiment L5 were used to drive the model, with $\delta$D adjusted by +10 ‰.

be used to identify and understand the physical processes associated with sublimation which would occur on a slower timescale in nature. Laboratory results show a strong signal of enrichment in the snow surface $\delta^{18}O$, as light isotopes preferentially sublimate from the surface due to fractionation. In addition we observe a strong decrease in the d-excess. This indicates that the HD$^{16}$O water isotopes are preferentially removed compared to H$_2^{18}$O. This is associated with the presence of a strong

humidity gradient leading to kinetic fractionation processes. We are able to reproduce these effects on the evolution of the snow isotopes using a simple box model based on isotope mass balance equations and the interstitial vapor diffusion model (Johnsen et al., 2000). This aligns with previous experimental and modeling studies (Ritter et al., 2016; Ebner et al., 2017), and confirms our hypothesis that the upper several centimeters of the snow surface are rapidly (i.e. on a sub-daily timescale) influenced by equilibrium and kinetic fractionation during sublimation. This contradicts the traditional theory of sublimation, which states

that sublimation occurs layer-by-layer and does not alter the snow isotopic composition, on which ice core paleoclimate water isotope research has been resting upon (Dansgaard et al., 1973).

In order to interpret these results in the context of natural processes, we consider the results of the field experiments. Previous studies have shown that significant isotopic changes of surface snow are observed (using daily sampling resolution) over periods of time without precipitation, and this is associated with snow metamorphism (Steen-Larsen et al., 2014; Casado et al., 2018).

We expand on these findings with higher resolution field sampling, showing that snow surface $\delta^{18}O$ and d-excess change on an hourly basis, which was hypothesized by Madsen et al. (2019); this demonstrates that similar processes to the lab experiments are occurring in a natural environment, albeit less extreme.

To interpret the driving factors in snow isotope changes, we consider differences between the FB box and FS surface samples. The FB samples were covered to shield from direct sunlight and wind-blown snow, and therefore were less likely to experience

vapor deposition or frost. Figure 8 shows a comparison of the top 0-0.5 cm sample for all FB and FS experiments with latent heat flux (LHF). Over the course of the field experiments, we observe several 6-12 hour periods of increasing $\delta^{18}O$ in 0-0.5 cm FB and FS samples, primarily during periods of positive LHF and decreasing d-excess; this is indicative of sublimation



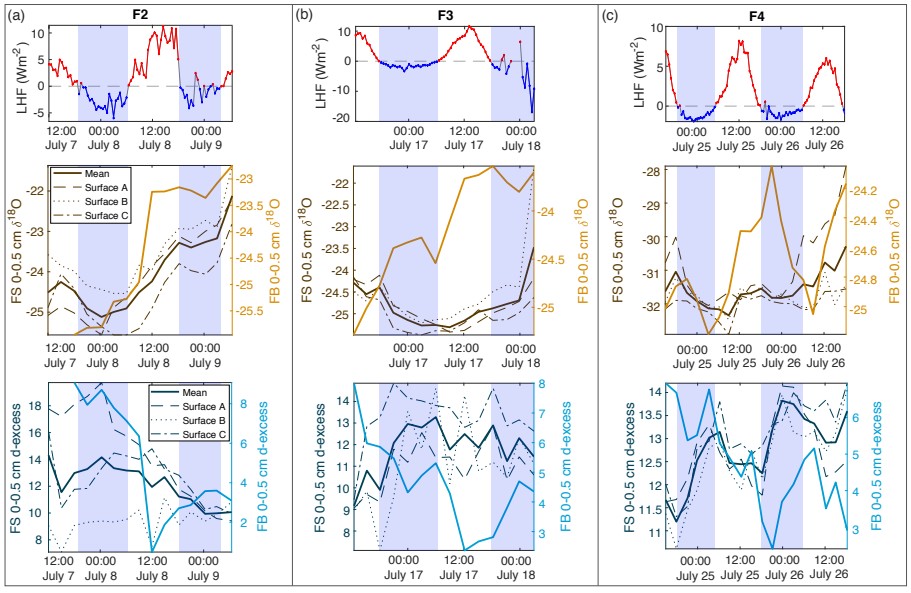

**Figure 8.** Comparison of latent heat flux (LHF) and 0-0.5 cm samples for mean FS surface samples and FB box samples for (a) F2, (b) F3, and (c) F4. Positive LHF values are indicated in red, and negative LHF values are marked blue with associated shading in all subplots (LHF, $\delta^{18}$O, and d-excess). FS surface snow 0-0.5 cm values are shown in dark orange ($\delta^{18}$O) and dark blue (d-excess), with each location (A, B, C) designated by dashed lines and the mean of all locations as the bold solid line. FB box snow 0-0.5 cm values are shown in light orange ($\delta^{18}$O) and light blue (d-excess) bold lines.

as suggested by laboratory experiments and model results. We also observe several 6-12 hour periods in which the FS $\delta^{18}$O decreases, despite experiments taking place during time periods with no precipitation and minimal wind-drifted snow. Periods

of decreasing FS $\delta^{18}$O occur primarily during nighttime hours with negative or low LHF measurements (Fig. 8; negative LHF indicated by shading) and increasing d-excess. There is no significant decrease in $\delta^{18}$O in FB2 and FB3 associated with these periods, while there is a simultaneous decrease in $\delta^{18}$O in FB4 and FS4. Additionally, the 0-0.5 cm d-excess decreases substantially in all FB experiments, similar to the signal that was observed due to kinetic fractionation during sublimation in laboratory experiments. In general, the box samples experience less decrease in $\delta^{18}$O than associated FS samples due

to minimized vapor deposition, and greater decrease in d-excess due to increased sublimation. This demonstrates that vapor deposition of preferentially isotopically heavy water molecules in the form of frost significantly contributes to the surface snow signal on a rapid timescale (Casado et al., 2018).

There are still several factors in the field experiments which could complicate interpretation of the results. While it is clear in the laboratory experiments that any changes in the snow composition are a direct result of sublimation, we cannot isolate

individual processes occurring in field experiments. For example, atmospheric vapor $\delta^{18}$O measurements often vary in phase with LHF, but during some periods (most notably, the latter half of F1 and F3) vapor $\delta^{18}$O deviates from the LHF trend. At this stage it is unclear whether LHF, vapor $\delta^{18}$O, or another factor is influencing the snow surface, or whether the snow surface





composition is driving vapor $\delta^{18}$O. Additionally, the isotopic composition of deeper snow layers could influence the surface snow due to diffusion. We note a general trend observed in Experiments FS1, FS2, and FS3 in which the deepest surface sample
(2-4 cm) has the lowest values for both $\delta^{18}$O and d-excess. However, throughout the duration of FS4, the upper samples (0-0.5, 0-1, and 1-2 cm) have a lower $\delta^{18}$O value than the 2-4 cm sample, likely due to a precipitation event preceding FS4 which may have deposited surface snow with anomalously low $\delta^{18}$O. If there are significant differences between the composition of adjacent snow layers, the surface snow could be influenced by a combination of interstitial diffusion and atmospheric driving forces (i.e. LHF and vapor $\delta^{18}$O). This may also explain some isotope inter-experiment differences between FB and FS results,
as FB samples are homogeneous and FS samples have vertical variability in snow isotopic composition.

    A key finding from field experiments is that both sublimation and vapor deposition influence the surface snow on an hourly timescale; this is supported by laboratory experiments and model results, demonstrating that sublimation has the ability to influence the mean surface snow isotopic composition in the top 2-3 cm of the snowpack during precipitation-free periods. These changes are occurring faster than the average recurrence of precipitation events, and could produce substantial changes in
the mean isotopic composition of the upper several cm of the snowpack over a long precipitation-free period. This suggests that effects from sublimation and vapor deposition may be superimposed on the precipitation signal, resulting in a snowpack record more indicative of atmospheric conditions and water vapor isotopic composition than condensation temperature (i.e. $\delta^{18}$O) or precipitation source region conditions (i.e. d-excess). The extent to which this occurs is dependent on the accumulation rate at the ice core site, as these processes primarily influence the top few cm of the snow column. A site such as Renland
(east-central Greenland), which receives 45 cm per year ice equivalent precipitation (i.e. several meters of snowfall), will be less affected than a drier location with significantly less annual accumulation, such as Antarctic sites like WAIS Divide (24 cm annual accumulation) (Fudge et al., 2016) or South Pole (7.4 cm annual accumulation) (Mosley-Thompson et al., 1999).

    To assess the relevance of our results in the context of ice core records, we compare the scale of the daily sublimation-driven changes observed in our experiments to the scale of the seasonal amplitude in the isotope signal. The Renland Ice Cap,
for example, has documented seasonal amplitude (i.e. summer peak to winter trough) of ∼8 ‰ $\delta^{18}$O (Hughes et al., 2020), and roughly 5-8 ‰ d-excess at the surface (i.e. relatively undiffused). In our laboratory experiments we observe changes of up to 8 ‰ $\delta^{18}$O and 20 ‰ d-excess over time periods of several days, and in FS field experiments we find up to 3 ‰ changes in $\delta^{18}$O and over 4 ‰ in d-excess on very short (sub-diurnal) timescales. This suggests that under typical natural conditions, changes in the mean value occurring on a short timescale could have a substantial impact on the seasonally-
recorded isotope signal. Previous studies have addressed the effect of seasonally-biased accumulation rate on diffusion and the recorded $\delta^{18}$O isotope signal (Persson et al., 2011; Casado et al., 2020; Hughes et al., 2020) and the effect of physical modifications and snow redistribution of the snow surface on the accumulation intermittency (Zuhr et al., 2021), but the effect of sublimation driven changes in surface snow isotopic composition between precipitation events has not been quantified previously. Whether the magnitude of the mean isotope change due to sublimation and snow-vapor exchange outweighs the
effects of snow redistribution, accumulation bias, and diffusion has yet to be determined.

    In order to fully understand the implications of sublimation and snow-vapor isotope exchange on the ice core record, it is necessary to quantify the effects of these processes over the course of a full year. While not in the scope of this paper,



this problem can first be approached through mass and isotope flux measurements throughout the summer field season (Wahl et al., In review). Subsequent modeling of these processes throughout the annual climate cycle will provide insight as to

what magnitude snow-vapor exchange influences surface snow on longer timescales (i.e. months to years), and how it may be recorded in the ice core isotope record. This could inform us to what extent changes in frequency of precipitation events, accumulation rate, and latent heat flux could influence the isotope signal recorded in ice cores on decadal-to-millennial scales. Our findings suggest that these variables contribute to a combined isotope signal, in which $\delta^{18}O$ and d-excess in ice core records likely incorporate individual precipitation events (i.e. condensation temperature and moisture source region conditions,

respectively), surface redistribution (i.e. wind drift and erosion), and a post-depositional alteration signal reflecting atmospheric conditions at the ice core site. Snow isotope models such as CROCUSiso (Touzeau et al., 2018), the Community Firn Model (Stevens et al., 2020), and isotope-enabled climate models would therefore be improved through the incorporation of isotope fractionation during sublimation, snow-vapor isotope exchange, and snow metamorphosis.

## 5    Conclusions

In this study, we have combined controlled laboratory experiments, field measurements, and model results, in an effort to constrain the effects of sublimation on surface snow isotopic composition. Experiments in a controlled laboratory setting demonstrate isotopic enrichment due to fractionation occurring during sublimation. We have reproduced laboratory results using a simple box model based on observed vapor measurements, a mass-balance equation and the Johnsen et al. (2000) vapor diffusion model. In both the experimental and model results, $\delta^{18}O$ increases as light isotopes preferentially sublimate due

to fractionation, and d-excess decreases due to kinetic fractionation. These changes occur rapidly, substantially changing the isotopic composition of the top 2-3 cm of snow over a 4-6 day period. Field experiments included continuous measurements of atmospheric vapor and latent heat flux during periods of high-resolution surface snow sampling, during which we observed significant changes in snow surface isotopes on a sub-diurnal timescale. We observed periods of increasing and decreasing $\delta^{18}O$, indicating that both sublimation and vapor deposition respectively influence the surface snow on an hourly basis. This

supports our hypothesis that rapid change occurs in a natural setting and propagates into the snowpack, substantially altering the initial precipitation isotope signal.

Post-depositional effects have implications for the interpretation of ice core data, which traditionally is assumed to only record isotopic variability from precipitation. Our results complement previous studies demonstrating spatial and temporal variability in snow surface isotopes, further strengthening the idea that the ice core record not only integrates the climate signal

of condensation temperature (i.e. $\delta^{18}O$ and $\delta D$) and moisture source conditions (i.e. d-excess) during precipitation, but also integrates the atmospheric conditions between precipitation events (in both $\delta^{18}O$ and d-excess). These factors should in the future be included in isotope-enabled climate models, which may include estimates of synoptic-scale patterns across annual cycles that would influence latent heat flux, vapor composition, and the resulting influence on surface snow isotopes. This will improve future interpretations of ice core data and may be the missing link in the transfer function between climate and an



uninterrupted isotope record, strengthening our interpretation of ice core water isotopes as a proxy for a continuous integrated

climate record.

*Data availability.*   All data has been submitted to PANGAEA and is currently under review. A DOI will be added when available.





**Appendix A: Figures and Tables**

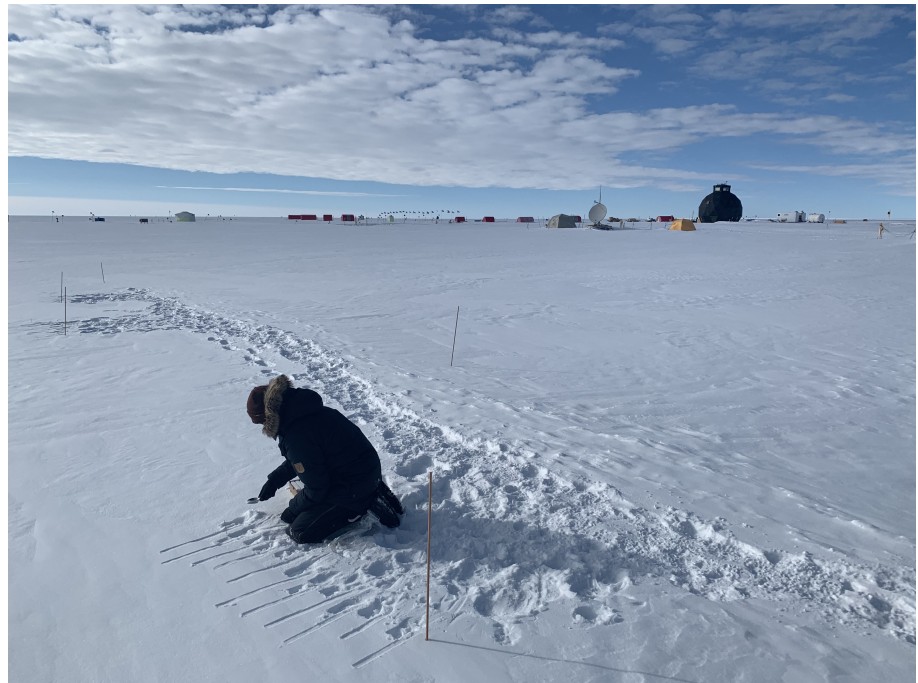

**Figure A1.** A photo of the FS sampling location shows the proximity between individual samples and sites. In the foreground is FS site A being sampled, with sites B and C seen in the background.





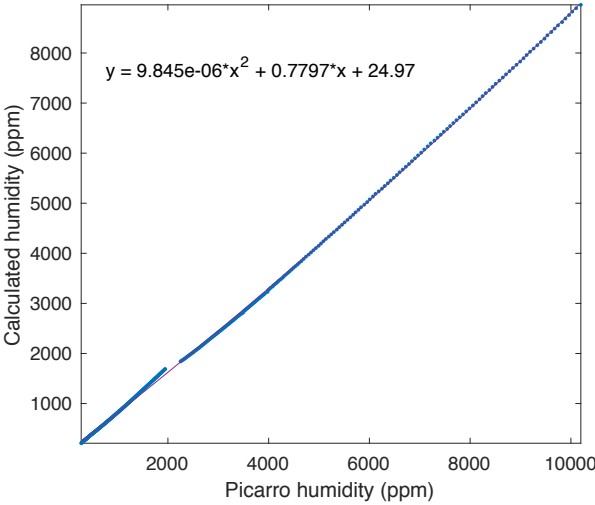

**Figure A2.** A comparison between measured and true humidity (determined from saturation temperature) yields a quadratic response, which is used to calibrate the measured humidity.

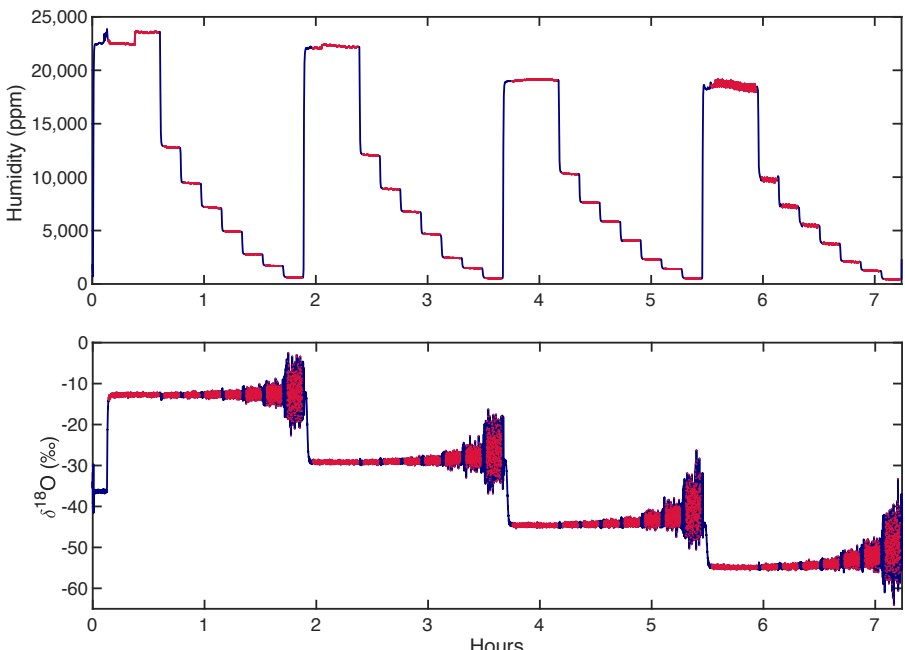

**Figure A3.** An example of a calibration sequence shows each standard (i.e. KBW -14.15 ‰, KAW -30.30 ‰, KPW -45.41 ‰, SPGW -55.18 ‰ $\delta^{18}$O) measured at multiple humidity levels for 12-20 minutes. The full calibration sequence (blue) is trimmed (red) such that the transition periods between humidity intervals and isotopic standards are ignored, and the average value of each trimmed period is calculated.

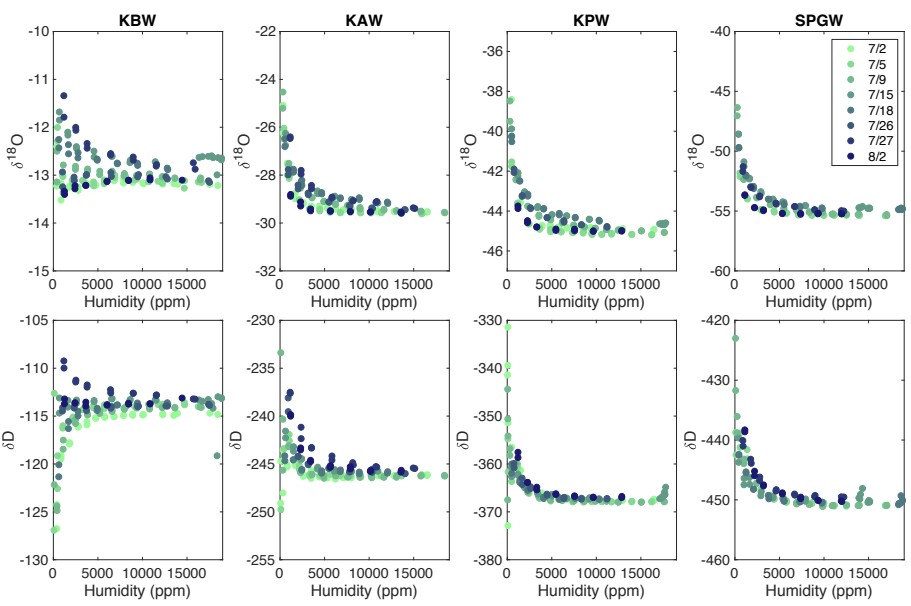

**Figure A4.** A compilation of all calibration runs from the 2019 EastGRIP field season demonstrates slight drift in the isotopic values, particularly at water concentrations less than 5,000 ppm. In total, eight calibration runs were completed (indicated by color) for four isotope standards (KBW -14.15 ‰, KAW -30.30 ‰, KPW -45.41 ‰, SPGW -55.18 ‰ $\delta^{18}$O; left to right, respectively).

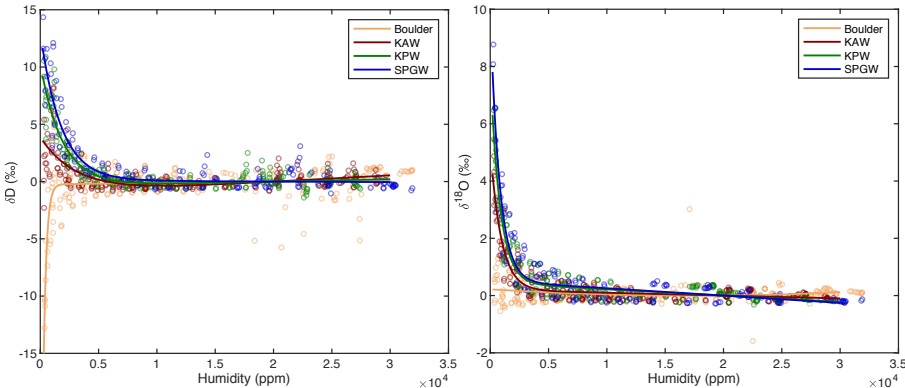

**Figure A5.** A double exponential curve is fit to the compiled calibration data for each standard for both $\delta^{18}$O and $\delta$D, characterizing the instrumental humidity-isotope response. This curve, normalized to the isotope value at 20,000 ppm, is used to correct the measured isotope data for bias at low humidity values.




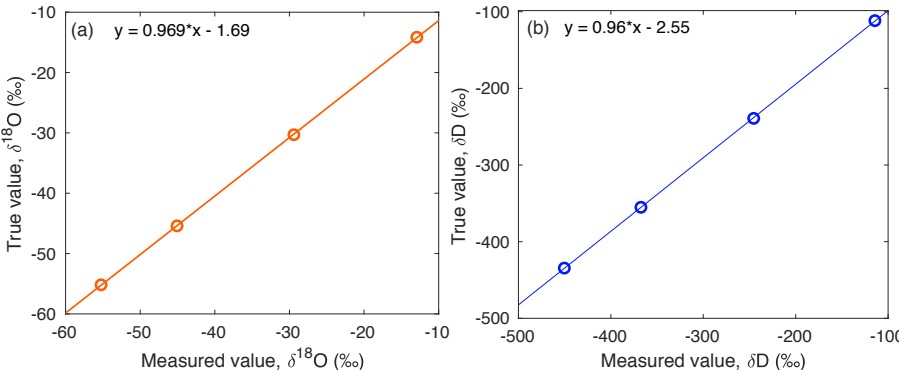

**Figure A6.** The true values of standards on the VSMOW-SLAP scale are compared to a compilation of standards measured across the field season, to yield a linear relationship for (a) $\delta^{18}$O and (b) d-excess. This correction is applied to measured isotope data.

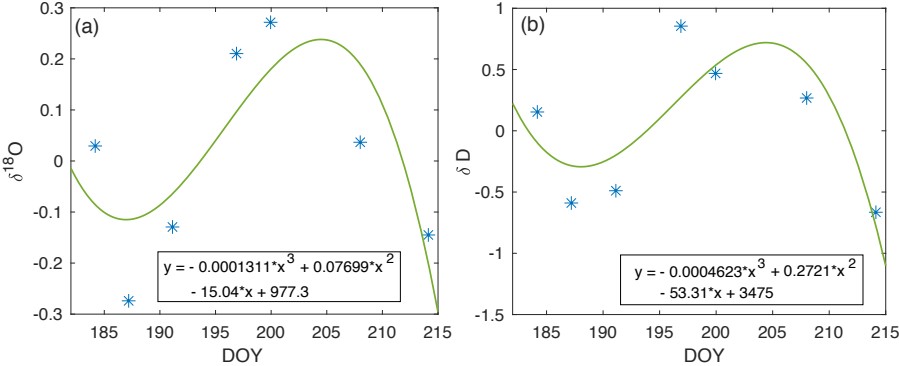

**Figure A7.** Drift in isotope measurements across the 2019 EGRIP field season (blue). The cubic polynomial curve fit (green) is used to correct experimental vapor measurements for the associated periods of time.

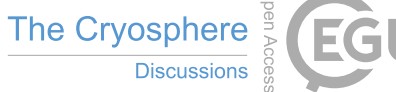



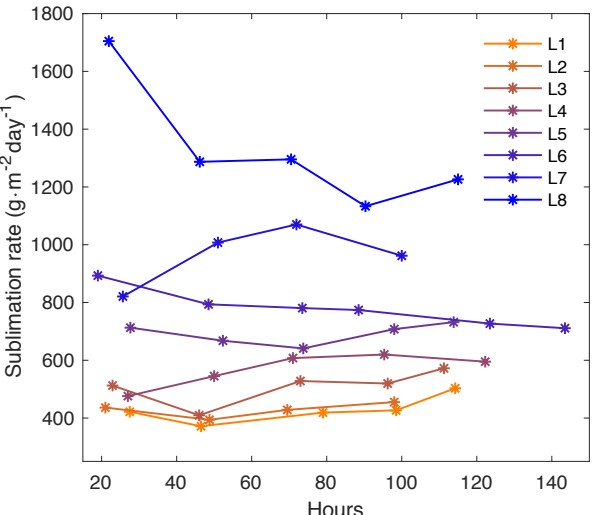

**Figure A8.** Sublimation rate with time for each laboratory experiment (L1-L8). The sublimation rate varies with temperature and dry air flow rate, and is relatively constant with time throughout each experiment.

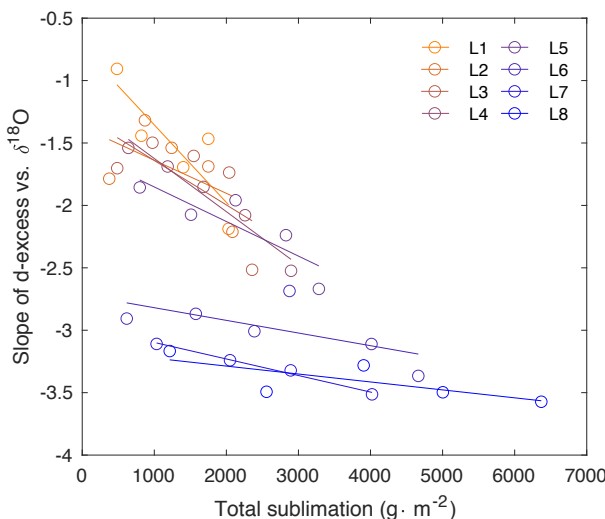

**Figure A9.** Slope of d-excess vs. $\delta^{18}O$ (as shown in Fig. 3) in comparison to total sublimation (sublimation rate * hours).





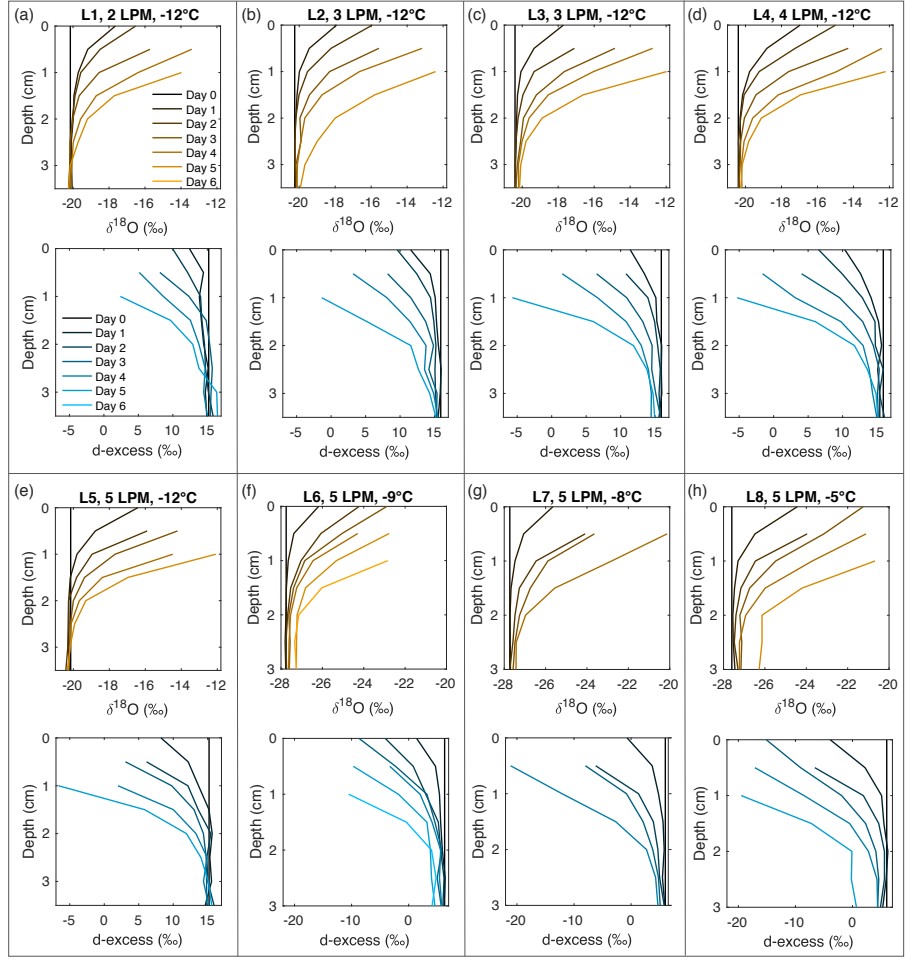

**Figure A10.** Snow $\delta^{18}$O (orange) and d-excess (blue) vertical profiles from all laboratory experiments (L1-L8; (a)-(h), respectively). Conditions for each experiment are indicated in each subplot. Day 0 (black) represents the initial homogeneous snow sample, with colors progressively moving towards orange ($\delta^{18}$O) and blue (d-excess) with each day of sampling. As each experiment progresses from Day 1 to Day 6, sublimation drives an increase in $\delta^{18}$O and decrease in d-excess, with the greatest change at the snow surface.

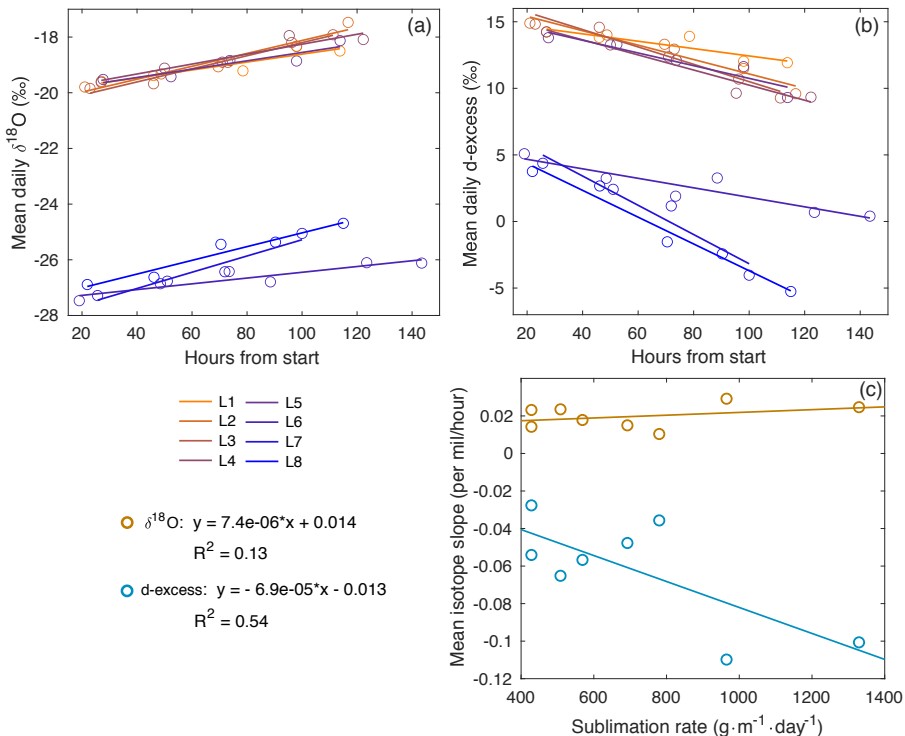

**Figure A11.** Mean daily (a) $\delta^{18}O$ and (b) d-excess with time for laboratory experiments L1-L8. The slope of the line for each experiment is represented in (c), compared to sublimation rate. There is a slight increase in $\delta^{18}O$ slope vs. sublimation rate ($R^2 = 0.13$), with a stronger relationship observed in the decrease in d-excess vs. sublimation rate ($R^2 = 0.54$).



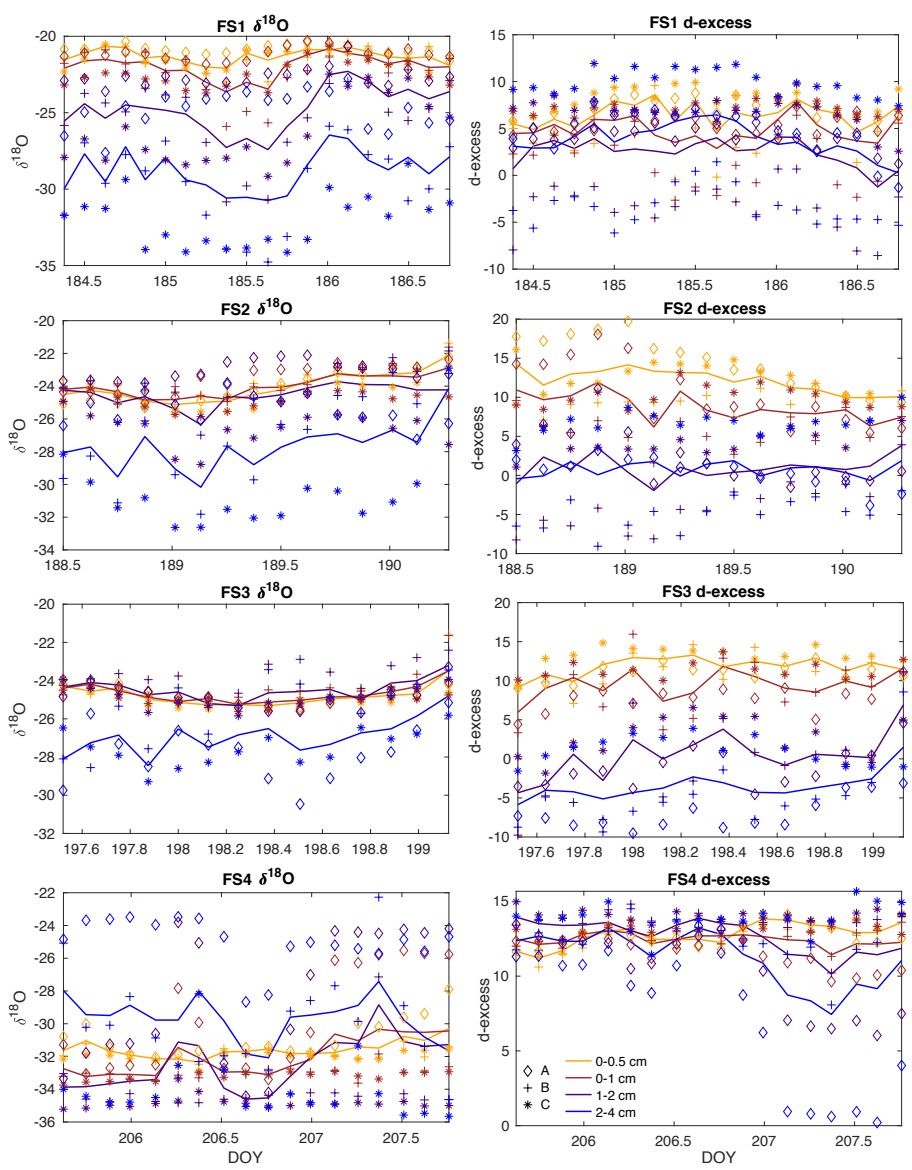

**Figure A12.** All surface samples are shown for field experiments FS1-FS4 (top to bottom, respectively), including $\delta^{18}$O (left column) and d-excess (right column). Symbols represent sampling locations (diamond, Site A; plus, Site B; asterisk, Site C), and colors indicate sampling height (yellow, 0-0.5 cm from surface; red, 0-1 cm; purple, 1-2 cm; blue, 2-4 cm). Solid lines are the average of the three sampling locations (A, B, C).




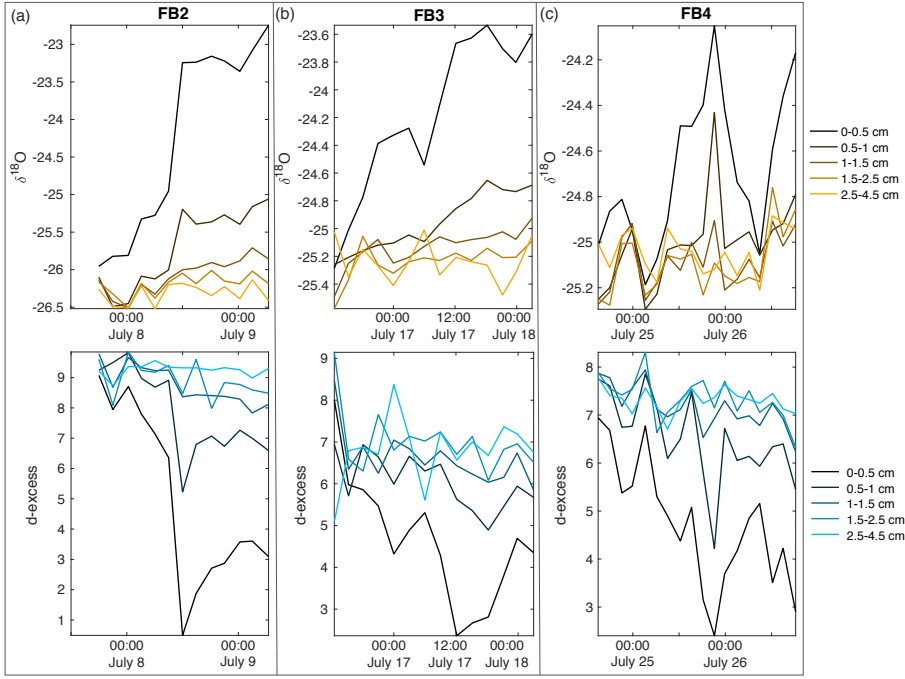

**Figure A13.** Field box samples are shown for (a) FB2, (b) FB3, and (c) FB4, including $\delta^{18}$O (top row) and d-excess (bottom row). Colors indicate sampling depth from surface; black is the surface sample from 0-0.5 cm, progressing with depth towards light orange ($\delta^{18}$O) and light blue (d-excess) at 2.5-4 cm below the surface.





**Figure A14.** Additional atmospheric conditions are shown for all field experiments F1-F4 (a-d, respectively). From top: latent heat flux (red, positive values; blue, negative values; dashed gray line at 0); temperature (orange); and atmospheric vapor measurements at 10 cm above the snow surface; humidity (purple), $\delta^{18}O$ (green), and d-excess (teal).





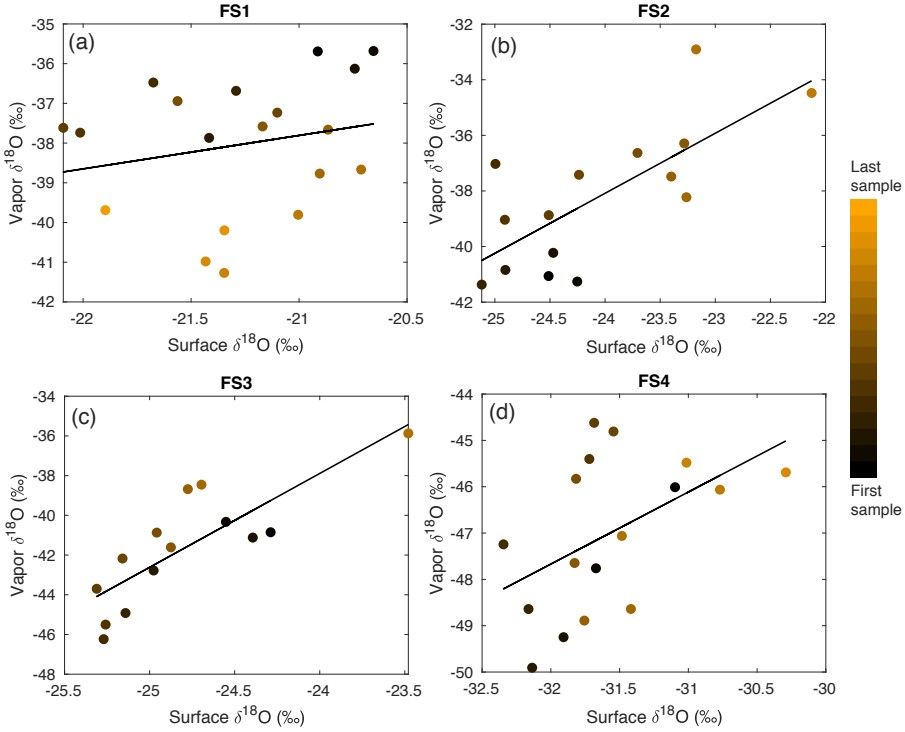

**Figure A15.** A comparison between $\delta^{18}$O of vapor and the top (0-0.5 cm) FS sample shows a significant relationship in FS2, FS3, and FS4, determined by P-values≤0.05. The sampling time is indicated by a color scale from black (first sample taken) to orange (last sample taken), and a linear regression is calculated for each experiment.





**Table A1.** Standards used in field and laboratory experiments.

| Standard | $\delta^{18}$O (‰) | $\delta$D (‰) |
|---|---|---|
| Boulder (KBW) | -14.15 | -111.65 |
| Antarctic (KAW) | -30.30 | -239.13 |
| Polar (KPW) | -45.41 | -355.18 |
| South Pole Glacial (SPGW) | -55.18 | -434.47 |
| Bermuda | -0.25 | 2.1 |
| NEEM | -33.50 | -257.1 |
| -40 | -39.93 | -310.7 |
| DC02 | -54.07 | -428.2 |





**Appendix B: Vapor calibrations**

The following four types of calibrations were performed to calibrate the water vapor isotope measurements of the CRDS, similar to the calibration protocol described in Steen-Larsen et al. (2013): 1) Humidity; 2) Humidity-isotope; 3) VSMOW-VSLAP; 4) Drift. For all isotope calibrations in both laboratory and field setups, the liquid standard was first vaporized using a nebulizer system, which produced vapor at 20,000-25,000 ppm. This vapor was combined with a dry air source using an open split, and a mass flow controller regulated the flow of dry air ranging from 10-21 cc·min$^{-1}$. The CRDS inlet constantly pulled a vacuum at 30 cc·min$^{-1}$; therefore, the remaining air flow is pulled from the humidified nebulizer source. This allowed for a constant stream of vapor at a controlled isotopic value and humidity level.

The calibration runs performed before and after each field experiment consisted of a seven-hour cycle with each of the four standards (Table A1) measured for 12 minutes at eight different humidity levels from 500-12000 ppm, as well as a half-hour measurement at >20,000 ppm (Fig. A3). This calibration was performed before and after each field experiment run (Fig. A4). Vapor measurements have uncertainty of 0.23 ‰ $\delta^{18}$O and 1.4 ‰ $\delta$D (Steen-Larsen et al., 2013). Details of each laboratory and field calibration type are as follows:

1. *Humidity calibration*: The measured humidity was corrected to the true humidity using a polynomial relationship, which was determined by calibration to a range of known humidity levels (Fig. A2). This instrument-specific relationship was not expected to significantly drift with time.

   (a) *Laboratory*: A laboratory humidity calibration was carried out by drawing humid air through a chilled ethanol bath, calculating the true humidity based on the saturation vapor pressure of the bath temperature (Fig. A2).

   (b) *Field*: The humidity calibration required a full laboratory setup, which was not available in the field. Due to instrument damage during shipping, the calibration could not be performed after the field season. Therefore, the humidity measurements were calibrated to a second Picarro L2130-*i* CRDS instrument which was continuously measuring atmospheric vapor ~30 m away and was calibrated for humidity. Simultaneous humidity measurements were matched and used to calibrate the CRDS instrument measurements reported here.

2. *Humidity-isotope response calibrations*: Isotopic bias occurs at lower humidity level (i.e. less than 10,000 ppm), and is sensitive to isotope concentration (Weng et al., 2020). Because experimental vapor measurements are typically below 5,000 ppm, it is important to perform a rigorous calibration of multiple isotopic standards (Table A1) at varying humidity levels (Figs. A3, A4). A double-exponential curve is fit to the isotope response with respect to humidity (Fig. A5), and is used to correct deviations in low-humidity experimental data.

   (a) *Laboratory*: The humidity-isotope response was determined by a full calibration of four isotopic standards (Table A1) measured for 20-30 minutes at a range of humidity levels from ~500-10,000 ppm. For experiments L1-L5, vapor measurements were calibrated to KAW, and vapor measurements in experiments L6-L8 were calibrated to NEEM. These standards were closest to the isotopic values of the vapor measurements, which differed between experiments due to different starting snow isotopic composition.





(b) *Field*: All calibration runs performed before and after each experiment run were compiled for the full field season. A double-exponential curve was fit to the compilation of data for each standard to determine the mean instrument response to humidity (Fig. A5). The vapor measurements were calibrated using the mean curve for KPW, which has the closest isotope value to average vapor measurements.

3. *VSMOW-SLAP scale calibration*: The calibration to the VSMOW-SLAP scale was established using standards (Table A1) which bracketed the measured water vapor isotope data (Fig. A6). A linear relationship is calculated between the 'True value', or the established isotopic standard value, and the 'Measured value', which is calculated from standard measurements.

   (a) *Laboratory*: The 'Measured values' for the VSMOW-VSLAP scale were taken from the isotopic values at ∼6,000-8,000 ppm measured in the full calibration used for the humidity-isotope response curve.

   (b) *Field*: The 'Measured values' were derived from the mean value of each standard measured for >10 min at 20,000-35,000 ppm over the course of the field season.

4. *Drift calibration*: While points 2) and 3) above account for mean instrument isotope deviations from standard values, instrument drift with time has also been observed in CRDS. For this reason, we calculated a best fit with respect to time for the isotope values at higher humidities from each calibration performed (Fig. A7). Deviations from the mean are then corrected for within each experiment period.

   (a) *Laboratory*: A short calibration of 3-4 standards at 3-4 humidity levels was completed before and after each experiment run. Instrument drift was calculated from the variability in KAW (L1-L5) and NEEM (L6-L8) at 2000 ppm.

   (b) *Field*: The isotope drift with respect to time was calculated from the mean value of measurements at 20,000 ppm for each calibration.

**Appendix C: Isotope model**

A simple box model is used to replicate the changes observed in the snow column throughout Experiment L5. First, a mass balance equation is used to derive the change in the surface snow isotopic composition with respect to time, due to sublimative mass loss. For a 10-second timestep $i$:

$$R_S^{i+1} = \frac{m^i R_S^i - dm R_E^i}{m^i - dm} \tag{C1}$$

Where $R_S$ is the isotopic composition of the snow surface (top 0.5 cm), and $R_E$ is the composition of the vapor as measured in Experiment L5 (see Eq. 1, Fig. 4). $m$ is the mass per unit surface area and $dm$ is the change (in this case, loss) of mass from





$i$ to $i+1$, such that $dm = sr \cdot dt$ for a given sublimation rate ($sr$) and time step ($dt$). Using the substitution $m^{i+1} = m^i - dm = m^i - (sr \cdot dt)$, we can find the partial derivative with respect to time:

$$\frac{\partial R_S}{\partial t} = \frac{sr}{m}(R_S - R_E) \tag{C2}$$

Which can be converted to $\delta$ notation using the relationship $R = \frac{\delta}{1000} + 1$:

$$\frac{\partial \delta_S}{\partial t} = \frac{sr}{m}(\delta_S - \delta_E) \tag{C3}$$

Equation C3 allows us to calculate the change in the isotopic composition of the snow surface directly from the isotopic composition of the sublimative flux. We can then model the propagation of the isotope signal through the snow column using the Johnsen et al. (2000) model of vapor diffusion in the firn. The diffusivity of water vapor in air is given by:

$$\Omega_a = 0.211 \left(\frac{T}{T_0}\right)^{1.94} \left(\frac{P_0}{P}\right) \tag{C4}$$

For temperature $T$, absolute temperature $T_0$, pressure $P$, and ambient pressure $P_0$. This is modified for an isotopic species $i$:

$$\Omega_{ai} = \frac{\Omega_a}{D_i} \tag{C5}$$

Where $D_i$ is the diffusivity of an isotopic species in air; $D_{18} = 1.0251$ and $D_D = 1.0285$. We can then calculate the diffusivity of an isotopic species in the firn:

$$\Omega_{fi} = \frac{mP\Omega_{ai}}{RT\alpha_i\tau}\left(\frac{1}{\rho_S} - \frac{1}{\rho_{ice}}\right) \tag{C6}$$

For molar mass of water $m$, density of snow $\rho_S$, density of ice $\rho_{ice}$, and tortuosity factor $\tau$:

$$\frac{1}{\tau} = 1 - 1.3 \cdot \left(\frac{\rho_S}{\rho_{ice}}\right) \tag{C7}$$

Finally, we have the diffusion equation for smoothing of an isotope profile:

$$\frac{\partial \delta}{\partial t} = \Omega_{fi}\frac{\partial^2 \delta}{\partial z^2} \tag{C8}$$

Because the height of the snow column is changing with time as a result of sublimation, we introduce the dimensionless 490    parameter $\xi$:

$$\xi = \frac{z}{H(t)} \tag{C9}$$





$$\partial z = H \cdot \partial \xi \tag{C10}$$

For the depth profile $z$ and the total height of the snow column $H$. This gives the diffusion equation as:

$$\frac{\partial \delta}{\partial t} = \frac{\Omega_{fi}}{H^2} \left( \frac{\partial^2 \delta}{\partial \xi^2} \right) \tag{C11}$$

We can now use the Forward Euler scheme to solve the diffusion equation, computing $\delta$ for time $i$ and depth $j$:

$$\frac{\delta_j^{i+1} - \delta_j^i}{\Delta t} = \frac{\Omega_{fi}}{H^2} \left( \frac{\delta_{j+1}^i - 2\delta_j^i + \delta_{j-1}^i}{\Delta \xi^2} \right) \tag{C12}$$

This can be computed using a matrix:

$$\delta_j^{i+1} = \delta_j^i + \frac{\Omega_{fi}}{H^2} \cdot \frac{1}{\Delta \xi^2} [A] \Delta t \tag{C13}$$

Where $[A]$ is the matrix :

$$A = \begin{bmatrix} -1 & 1 & 0 & \cdots & & & & 0 \\ 0 & 1 & -2 & 1 & 0 & \cdots & & 0 \\ 0 & \cdots & 1 & -2 & 1 & 0 & \cdots & 0 \\ \vdots & & & \ddots & & & & \vdots \\ 0 & \cdots & & & & & 1 & -1 \end{bmatrix} \tag{C14}$$

The first and last rows set boundary layers such that there is no diffusion in or out of the top or bottom layers of the snow column. We can write out Equation C3 in a similar manner:

$$\delta_j^{i+1} = \delta_j^i + \frac{sr}{m} (\delta_S - \delta_E)[B] \Delta t \tag{C15}$$

$$B = \begin{bmatrix} 1 \\ 0 \\ \vdots \\ 0 \end{bmatrix} \tag{C16}$$





Where the matrix $[B]$ represents only the isotope change in the surface. Therefore, combining Equations C13 and C15, we have the following:

$$\delta_j^{i+1} = \delta_j^i + \frac{\Omega_{fi}}{H^2} \cdot \frac{1}{\Delta\xi^2}[A]\Delta t + \frac{sr}{m}(\delta_S - \delta_E)[B]\Delta t \qquad \text{(C17)}$$

Thus, Eq. C17 can be used in conjunction with Eq. C1 to model the evolution of the snow column with respect to time,
driven by $R_E$, the composition of the measured atmospheric vapor.



*Author contributions.* AGH and HCSL designed the laboratory and field setup and experiments. AGH carried out the experiments with significant contributions from HCSL. SW and AZ assisted with field snow sampling, MH assisted with materials and production ideas of the laboratory experimental chamber. AGH wrote the article with significant contributions from HCSL, TRJ, and SW, and edits from all authors.

*Competing interests.* The authors declare that they have no conflict of interest.

*Acknowledgements.* Abigail Hughes acknowledges support from the National Science Foundation through the Graduate Research Fellowship Program (grant No. DGE 1650115). This manuscript has received funding from the European Research Council (ERC) under the European Union's Horizon 2020 research and innovation program: Starting Grant SNOWISO (grant agreement No. 759526). EGRIP is directed and organized by the Centre for Ice and Climate at the Niels Bohr Institute, University of Copenhagen. It is supported by funding agencies and institutions in Denmark (A. P. Møller Foundation, University of Copenhagen), USA (US National Science Foundation, Office of Polar Pro-
grams), Germany (Alfred Wegener Institute, Helmholtz Centre for Polar and Marine Research), Japan (National Institute of Polar Research and Arctic Challenge for Sustainability), Norway (University of Bergen and Trond Mohn Foundation), Switzerland (Swiss National Science Foundation), France (French Polar Institute Paul-Emile Victor, Institute for Geosciences and Environmental research), Canada (University of Manitoba) and China (Chinese Academy of Sciences and Beijing Normal University). Funding has also been provided by the National Science Foundation program of Arctic Natural Sciences (grant No. 1804098). The authors also thank the Stable Isotope Lab at the University
of Colorado Boulder for facility and instrument use, and Bruce Vaughn and Valerie Morris for assistance.



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
