# Peer review of "The role of sublimation as a driver of climate signals in the water isotope content of surface snow: Laboratory and field experimental results"

_The Cryosphere, 2021_

## Referee Comment (RC3)

[referee-annotated manuscript omitted]

---

## Author Comment (AC1)

RC1

Review of manuscript "The role of sublimation as a driver of climate signals in the water isotope content of surface snow: Laboratory and field experimental results" by Abigail Hughes and others.

This work is devoted to the investigation of the post-depositional changes of the snow isotopic composition due to the mass- and isotopic exchange between snow cover and the overlying atmospheric water vapor. The authors use the results of laboratory experiments, as well as of two types of filed experiments, to show that the isotopic composition of the uppermost few cm of snow may change at hourly time-scale due to these processes. The obtained results are quite interesting and important as another step towards a comprehensive transfer function between isotopic content of precipitation and that of an ice core.

I have a few minor comments and questions as listed below:
Figure 1 - photo of the experimental set-up would be relevant.

Photos of the experimental setup have been added to Appendix A.

Lines 111-112: "Three Pico Technologies PT-104 Data Logger temperature sensors were placed in the box to record continuously; one 10 cm above the snow surface" – based on the figure 1, the upper sensor is placed about 20 cm above the snow surface.

The height of the sensor in Figure 1 has been adjusted to be better to scale at 10 cm above the snow surface.

Line 134: "wind speeds below 10-12 knots" – meters per second is a preferable dimension in meteorology.

We have changed "10-12 knots" to 5-6 m/s"

Lines 136-137: "Sampling boxes were partially buried in the snow surface, and protected from direct overhead sunlight using a cloth covering" – why did not you bury the boxes completely in the snow (so that the level of snow in and around the box is the same), and why did not you use a white (a sunlight-reflecting) material for the box?

If the level of snow surrounding the box was at top of the box, the risk of wind-blown snow grains contaminating the samples may have been higher. While we completed experiments under low wind conditions to mitigate this risk, we chose to only partially bury the boxes to keep the risk of contamination as low as possible. While white boxes may have reflected sunlight better and could certainly be used in future follow-up experiments, the clear boxes were what we had available in the field.

Line 139: "which would have otherwise led to melt of the snow not otherwise occurring" – this sounds a bit awkward to me, please consider rephrasing.

"Which would have otherwise led to melt of the snow not otherwise occurring" has been changed to "which may have led to melt of the snow samples".

---

## Author Comment (AC2)

RC2

General comments:

This paper examined the role of sublimation as a driver of isotope climate signal preserved in ice cores. This study conducted two experiments in laboratory and field (Greenland) and a modelling for lab experiment. Each of the three major topics, experiment, modeling, and field experiments, are interesting and the data will be valuable. However, I think that each topic ends up with insufficient discussion/interpretation. In addition, because of the three topics, it is difficult to understand the main findings of this study. Please see my comments for details. Overall, substantial revision is needed for this MS.

Major comments

(1) L310 "This contradicts the traditional theory of sublimation, which states that sublimation occurs layer-by-layer and does not alter the snow isotopic composition….(Dansgaard et al., 1973)".

>This is one of major arguments of this paper. Please explain why the traditional theory is wrong. During sublimation, the remaining ice is not mixed. Thus, the isotope ratio is not controlled by typical Rayleigh distillation. This basic concept sounds very reasonable and therefore many people believe it.

The traditional theory postulates that the ice core water isotope signal is governed by the precipitation isotope effect. However due to the practical difficulties of sampling the snow fall on top of the Greenland and Antarctic Ice Sheets and the time scales involved in creating the ice core signal in the real world it has not been possible to establish the physical processes linking the snowfall water isotope and ice core isotope climate signal. However, previous work (Ritter et al., 2016, Casado et al., 2018, Steen-Larsen et al., 2014) has documented both for Antarctica and Greenland that the water stable isotope signal of the snow is influenced by post-depositional processes taking place between precipitation events. Thus, while the isotope signal of precipitation may be controlled by Rayleigh distillation, the recorded signal in the ice sheet reflects a combination of Rayleigh distillation and post-depositional processes.

Furthermore, previous laboratory experiments (i.e. Ebner et al., 2017; Sokratov and Golubev, 2009) have established that sublimation does influence the isotopic composition of remaining snow, and sublimation does not occur layer-by-layer as was traditionally thought. These previous laboratory and field experiments show that there are deficiencies in the traditional theory. Our manuscript aims to address these issues and work towards determining the extent to which the isotope signal in the ice core reflects precipitation (i.e. Rayleigh distillation), sublimation, and condensation.

(2) The model used in this paper is based on a mass balance, in which input data is the observed vapor isotope ratio. Thus, it is not surprising that the model result agrees with the observation. I think different approach is needed to understand the physical process behind this experiment. For example, Figure 4 shows significant decrease of relative humidity. This should affect kinetic fractionation during sublimation. Thus, the net isotope fractionation factor had changed during the experiment. How much this affects your observation? The impact of changing fractionation factor may be evaluated (i.e., Craig-Gordon model). This is important because the snow/vapor isotope composition changed because of changing fractionation factors.

The decreasing relative humidity does likely cause the fractionation factor to change; however, this is not the reason that the snow/vapor composition is changing. As lighter isotopes preferentially sublimate, the remaining fraction of snow becomes isotopically heavier. Subsequently, continuous sublimation causes both the remaining fraction of snow and the sublimated vapor to become isotopically heavier with time. While the changing fractionation factor may cause the relationship between the snow and sublimated vapor to vary slightly over the course of the experiment, the primary cause of changing snow and vapor composition is the removal of relatively more light isotopes than heavy isotopes from the system.

Because the removal of light isotopes drives the changing snow surface composition, it is reasonable to use a mass balance model to describe this physical process. Rather than calculating the vapor isotopic composition with a parameterization that was developed for water bodies (i.e. Craig-Gordon model), we therefore make use of the actual, measured vapor isotopic composition. The development of a full parameterization for the sublimation fractionation is beyond the scope of this paper, but could be developed in future modeling studies.

Furthermore, it is important to note that while we use a mass balance model to estimate changes in the snow surface (and a future full parameterization could do the same), only by including the Johnsen diffusion model are we able to simulate the full snow column. Thus, our model demonstrates not only that the isotopic composition of the snow surface changes due to sublimation, but that these changes can rapidly propagate deeper below the surface layer.

Additionally, we had explored utilizing the Craig-Gordon model to describe our experimental results, but found that we were not able to fit the model to our observations. This indicates that either the Craig-Gordon model for evaporation is not applicable to sublimation, or that it is not applicable to the rate of sublimation flux observed in the laboratory experiments. Thus, the mass balance model is more effective in describing our observations. That being said, we will note that fitting the values for k, n, or theta (as used in the Craig-Gordon model) would only be valid for the very specific environmental conditions used in the experiment, and a calculated kinetic fractionation factor would not be applicable to a real-world scenario. Thus, providing this number as a result may only introduce confusion to the reader.

(3) Please state clearly what is new compared to previous studies in Introduction. Sampling depths apper to be finer for this study? What new for the laboratory experiment?

The key novelty of these laboratory experiments compared to previous studies is the combination of continuous water vapor isotope measurements of the sublimated snow along with high resolution sampling of the snow pack for both d18O and dD measurements. The continuous water vapor isotope measurements allow us the possibility to directly measure the isotopic composition of the sublimate flux from the snow surface.

Previous laboratory studies only measured d18O (i.e. Ebner et. al., 2017), or only measured snow (i.e. Sokratov and Golubev, 2009). Previous field studies had lower sampling resolution (i.e. Steen-Larsen et al., 2014), or only measured surface snow (i.e. Casado et al., 2018, Ritter et al., 2016). Building off of these previous studies, our laboratory experiment is the first to measure d18O and dD at finer resolution of snow samples with continuous vapor measurements of d18O and dD. Our field experiments are the first to measure snow d18O and dD at multiple depths at hourly resolution for multiple multi-day experimental periods, with simultaneous continuous vapor measurements of d18O and dD.

To clarify this in the manuscript, we have added the following in L70: "Thus, these laboratory and field experiments are the first to measure both d18O and dD at fine vertical and temporal resolution for multiple depths across several multi-day experimental periods under differing environmental conditions, with simultaneous continuous measurements of atmospheric vapor d18O and dD. In case of the laboratory experiments presented, the vapor isotopic composition can directly be interpreted as the isotopic composition of the flux, since the experimental set-up fulfills the closure assumption and therefore allows a direct comparison of flux and snow isotopic composition. "

(4) Maybe the originality of this study is that the combination of the tree topics (lab, field, and model). If so, what did you "learn" from the combination? As the authors themselves noted, the laboratory experiment is difficult to compare the field result because of extreme condition of the lab experiment.

The benefit of utilizing multiple experiment types is that we are able to better interpret field observations. By isolating the process of sublimation in the laboratory experiments, we can determine the effect of sublimation on the snowpack. Thus, in the field experiments, we can differentiate between sublimation and other processes (such as condensation) that are occurring throughout the experimental period. While the effect of sublimation in the field is dampened compared to the laboratory conditions, we are able to identify time periods when it may be occurring (as discussed in Section 4).

(5) Reproducibility is the crucial for such experiment. Thus, please describe the setting of the experiment strictly (please refer to specific comments).

See responses to specific comments.

(6) I do not understand the exact purpose of the FB experiment because the condition (the box and cloth cover) is too far from nature. Maybe this is designed as an intermediate between laboratory and field?

Yes, the FB experiment is intended to be an intermediate between the laboratory and field. The FS samples, taken directly from the snow surface, are subject to variability in the snow surface isotopic composition due to wind blown snow, dunes, etc. For this reason we have three FS sample sites and report the average values, but it is also useful to include observations from FB snow samples that are subject to similar atmospheric conditions but are homogeneous at the start of the experiment.

(7) Please add the raw data and modelling code you used in supplementary material so that the readers can reproduce the figures.

The raw data is currently under revision for publication through PANGAEA and a DOI will be added when available. The reader would be able to reproduce figures using the model equations described in Appendix C.

Specific comments

L6 "how vapor-snow exchange and sublimation processes…."

>I think that the physical mechanism behind the vapor-snow exchange process is sublimation and deposition. Why did you say "the vapor-snow exchange and sublimation"? In fact, the two terms appeared several times throughout the paper.

To simplify this we have changed the following instances:

L6 from "vapor-snow exchange and sublimation" to "vapor-snow exchange"

L369 from "sublimation and snow-vapor exchange" to "sublimation and vapor deposition"

L371 from "sublimation and snow-vapor isotope exchange" to "sublimation and vapor deposition"

L.18 "our results demonstrate that post-depositional processes such as sublimation play a role…"

>Please clarify the difference from the previous findings.

L18-19 has been modified: "...our results demonstrate that post-depositional processes such as sublimation play a role in creating the climate signal recorded in the water isotopes in surface snow, both in laboratory and field settings." to clarify that we use findings from both laboratory and field experiments, which differs from previous studies.

L92 "Plywood box" > Add thickness.

L92 has been revised: "outer plywood box (2.7 cm thick)"

L.93 "PID-controlled heater" > Add details (e.g., what kind of heater (cable or panel)? Wattage?). Please also illustrate the location of the heater in Fig.1a.

L93 has been revised to "PID-controlled (Omega CN7800; 50 W) cable heater wrapped around the inside of the plywood box."

L.93 "with a generator" > Add details (product name, manufacture name etc.).

L93 has been revised to "with a generator (Puregas CDA-10)"

L94 "mass flow controller" > Add details (product name, manufacture name etc.). Please also illustrate this in Fig. 1a.

"...mass flow controller…" has been changed to "...HORIBA SEC-4400 mass flow controller…". To maintain simplicity in the diagram we have not changed the illustration, but the caption has been changed: "...dry air was pumped into the inner box…" has been changed to "...dry air regulated by a mass flow controller was pumped into the inner box…".

L95. "continuously-running fans" > Add details (wind speed, product name, manufacture name etc.). How many fans exactly did you install?

"...continuously-running fans maintained mixed air in the chamber" has been changed to "...two continuously-running computer fans at the top of the chamber maintained mixed air"

L.96 "small boxes" > Please add material used for this box. Please also add thickness of the boxes.

"...small boxes…" has been changed to "...small plastic boxes…". The thickness of the boxes is not relevant; the size described "(5.7x5.7x7.6 cm)" are the inner measurements of the box.

L.115 "experiments used snow"ã    > Please add details (type of snow, density etc.)

Detail added to L97: "...well-mixed and sifted so that snow grain size (1-2 mm) and isotope value was homogeneous…"

L.137 "partially buried" > How deep exactly?

L137 has been clarified: "Sampling boxes were partially buried in the snow surface, with the top of the sample box 1-2 cm above the surrounding snow surface to minimize any risk of contamination from wind-blown snow. Samples were also protected from direct overhead…"

L.137 "a cloth covering" > Please add details (material used, thickness, color). I do not understand why you used the cloth. Maybe the melting occurred because the boxes only "partially" buried?

L137 has been clarified: "...overhead sunlight using a light-colored thin cloth covering." The cloth is also visible in the photos in Fig. 1. Having the boxes partially buried may have contributed to melt, but it would have been very difficult to fully bury the boxes without risking contamination from the surrounding snow.

L.161 "KNF pump" > please add details (product name).

L161 has been revised: "...diaphragm pump (KNF model DC-B 12V UNMP850)..."

L.304 "..a strong decrease in the d-excess. This indicates that the HD16O water isotopes are preferentially removed compared to H218O"

> A decrease in the d-excess does not necessarily indicate that the HD16O is preferentially removed compared to H218O (i.e., HD16O are almost always preferentially removed compared to H218O because of larger isotope effect). The change of d-excess depends on changes in dD and d18O relative to slope of 8.

L303-305 have been revised: "In addition we observe a strong decrease in the snow surface d-excess. Decreasing d-excess driven by kinetic fractionation is also observed when a body of water evaporates into a sub-saturated atmosphere (Benson and White 1994, Merlivat and Jouzel 1979). As a similar effect is observed during sublimation in laboratory experiments, we draw the analogy that this is due to kinetic fractionation ."

L.304 "this indicates that the HD16O water isotopes are preferentially removed compared to H218O" > Precisely, the HD16O is an isotopologue of water. Furthermore, there is NO water isotopes, only oxygen (or hydrogen) isotope exists. But I know that many people used this term, "water isotope". Thus, it is not necessarily to revise "water isotope" throughout this manuscript. But this sentence is a bit strange.

In the instance of L304 "isotopes" is changed to "isotopologues" as we are specifically referencing HD16O. In other cases we prefer to use "water isotopes" which is commonly used throughout ice core literature.

L355 "A site such as Renland (east-central Greenland), which receives 45 cm per year …will be less affected…. "

> The SE-Dome core is a more suitable example of a high-accumulation site, which receives 102 cm per year. Furthermore, the ice-core d18O record is remarkably similar to the isotope-GCM outputs, suggesting negligible influence of post-depositional effect (Furukawa et al., 2017).

The example in L355 has been changed: "A site such as SE-Dome (southeast Greenland), which receives 102 cm per year ice equivalent precipitation (Furukawa et al., 2017) (i.e. several meters of snowfall), will be less affected…"

Reference: Furukawa, R., Uemura, R., Fujita, K., Sjolte, J., Yoshimura, K., Matoba, S., & Iizuka Y. (2017). Seasonal-scale dating of a shallow ice core from Greenland using oxygen isotope matching between data and simulation. Journal of Geophysical Research: Atmospheres, 122, 10,873 – 10,887. https://doi.org/10.1002/2017JD026716
Citation: https://doi.org/10.5194/tc-2021-87-RC2

---

## Author Comment (AC3)

**RC3**

General Comments:

The new, impressive, and labor intensive field measurements from East Greenland on snow-water vapor exchange, as well as laboratory measurements, are a valuable dataset which provides much needed insights on the effects of sublimation on the isotopic content of surface snow, and constraints on the post-depositional processes affecting the evolution of the isotopic composition of surface snow and atmosphere water vapor. The findings clearly show that sublimation does indeed impart an isotopic signal to the surface snow that propagates downward 1-2 cm over the period of 4-6 days during periods of clear skies. A simple box model is utilized to help understand the relatively isotopically enriched surface snow due to solid-vapor phase changes (i.e., sublimation) for d18O and the concomitant decrease in dexcess. The box model helps to understand and explain the combined effects of surface sublimation and diffusion of the signal observed at depth in the homogenous lab-based snowpack.

The comparison between field samples, field box samples, and laboratory experiments provides a good test of how large an isotopic effect sublimation has in a controlled environment (albeit extremely dry compared to field conditions). It also helps to identify certain environmental parameters that may be causing unexpected changes in the isotopic composition of field samples, like synoptic whether variations altering atmospheric vapor d18O over hourly timescales or if the snow surface composition is driving the vapor d18O due to deeper snow layers influencing the surface snow by internal diffusion between grains.

The authors highlight one of the key findings: "A key finding from field experiments is that both sublimation and vapor deposition influence the surface snow on an hourly timescale; this is supported by laboratory experiments and model results, demonstrating that sublimation has the ability to influence the mean surface snow isotopic composition in the top 2-3 cm of the snowpack during precipitation-free periods. These changes are occurring faster than the average recurrence of precipitation events, and could produce substantial changes in the mean isotopic composition of the upper several cm of the snowpack over a long precipitation-free period. This suggests that effects from sublimation and vapor deposition may be superimposed on the precipitation signal, resulting in a snowpack record more indicative of atmospheric conditions and water vapor isotopic composition than condensation temperature (i.e. d18O ) or precipitation source region conditions (i.e. d-excess). The extent to which this occurs is dependent on the accumulation rate at the ice core site, as these processes primarily influence the top few cm of the snow column."

Although the authors do an excellent job documenting the hourly changes to the surface snowpack in the field sample experiments (FS1-4), the question remains: what is the net effect to the snow pack isotopic composition over a weekly or monthly time period that is precipitation free? If sublimation enriches the snow surface and thus the overlying vapor isotopic composition, and then at night during negative LHF, equilibrium fractionation during vapor deposition should redeposit more δ18O negative water vapor onto the snow surface, and thus the net change is a minimal enrichment over weekly periods. For example, the initial FS2 0-0.5 cm values shown in Fig. 5 appear to have a mean value of ~−24 ‰ on July 7th and by July 9th the mean value is ~−22 ‰, smaller in magnitude than the δ18O increases observed in the FB 2-4 samples that were shaded/covered. However, by July 18th the mean value is −23.5 ‰ and then drops down to −32 ‰ ("likely due to a precipitation event preceding FS4 which may have deposited surface snow with anomalously lowd18O"), but the point is that over the ~3 week period the sublimation signal that should be slowly increasing δ18O is overwhelmed by either fresh precipitation or more depleted δ18O atmospheric water vapor from elsewhere. The sublimation of the surface snow on day-today timescales appears to be less important to the overall seasonal isotopic composition of the surface snow in regions like Greenland with more frequent synoptic systems and advection of water vapor from marine sources (e.g. Baffin Bay, Arctic Ocean or North Atlantic).

On the other hand the laboratory experiments are excellent demonstrators of intense sublimation over prolonged periods (using a continuous LHF equivalent to the max daily LHF in the field) and use a humidity level about 30-40x less than the atmospheric values found during the field experiments, which produces a very strong sublimation signal in the surface snow for L1-L8. The extreme sublimation rates make it a bit harder to draw comparisons to the field experiments, but provides an upper bound for the impact on isotopic enrichment of surface snow d18O and dexcess depletion during summer months. The smaller impact in the FS experiments shows that sublimation is still an important factor on diurnal timescales (daytime vs nighttime), but it remains unclear what the cumulative impact would be on the snowpack isotopic composition if at the end of the summer season a snow pit was sampled at 1cm increments would the sublimation changes be detectable or swamped by other post-depositional process (wind redeposition), synoptic scale atmospheric vapor imprints, or new precipitation events bringing in low d18O snowfall?  Clearly sublimation (and vapor deposition) is an important factor on the diurnal timescales during accumulation intermittency, but as the authors acknowledge: "Whether the magnitude of the mean isotope change due to sublimation and snow-vapor exchange outweighs the effects of snow redistribution, accumulation bias, and diffusion has yet to be determined."

The authors make a strong case that sublimation/vapor deposition changes do occur to the surface snow pack (~1-2cm for the FS snow surface samples) on sub-diurnal timescales. They argue that "These changes are occurring faster than the average recurrence of precipitation events, and [therefore] could produce substantial changes in the mean isotopic composition of the upper several cm of the snowpack over a long precipitation-free period."The authors then speculate that effects from sublimation and vapor deposition MAY be superimposed on the precipitation signal, "resulting in a snowpack record more indicative of atmospheric conditions and water vapor isotopic composition than condensation temperature (i.e.d18O) or precipitation source region conditions (i.e. d-excess)."  Although this is a reasonable speculation their data is not sufficient to support such conclusions about the monthly or seasonal timescale impacts of the two-way exchange driven by sublimation/condensation. Thus, it is not appropriate to for them to assess the relevance of their results to the scale of the seasonal amplitude in the isotope signal for a firn core from the Renland Ice Cap. The changes observed in the FS field experiments (mean of ~1.8‰ for FS1-4 d18O range based on Table 2) occur on short (multi-day) timescales but in order to compare the cumulative impact of these processes to the seasonal amplitude, they would need to have sampled a nearby snow pit at the start of the field campaign in early July and again at the end of the month to determine the net effect, and ideally throughout the entire summer (apparently new data will be available from Wahl et al., in review). The authors do acknowledge that "In order to fully understand the implications of sublimation and snow-vapor isotope exchange on the ice core record, it is necessary to quantify the effects of these processes over the course of a full year"and while that is not within the scope of this paper they go on to make concluding statements that the results support their hypothesis "that rapid change occurs in a natural setting and propagates into the snowpack, substantially altering the initial precipitation isotope signal."Although true on short timescales (sub-diurnal to diurnal) the results do not provide enough information to make definitive conclusions about the relative magnitude of sublimation/vapor deposition on longer timescales (i.e. years to decades) relevant to ice core interpretations.

I agree strongly with the authors that further research is needed over seasonal and annual timescales and that their results "suggest that these variables contribute to a combined isotope signal, in whichd18O and d-excess in ice core records likely incorporate individual precipitation

events (i.e. condensation temperature and moisture source region conditions, respectively), surface redistribution (i.e. wind drift and erosion), and a post-depositional alteration signal reflecting atmospheric conditions at the ice core site."Their suggestion that "Snow isotope models such as CROCUSiso (Touzeau et al., 2018), the Community Firn Model (Stevens et al., 2020), and isotope-enabled climate models" would therefore be improved through the incorporation of isotope fractionation during sublimation, snow-vapor isotope exchange, and snow metamorphosis." is certainly justified by their findings from both the laboratory and field experiments and results from such modeling efforts may help to interpret the relative contributions of the aforementioned processes affecting post-depositional changes.

Based on the above assessment, I would recommend acceptance with minor revisions but with a primary focus on revising the Discussion and Conclusion sections regarding the broader application of their findings to seasonal and yearly timescales, speculation on the cumulative effect (monthly, seasonal, or yearly) of short-term sublimation/vapor deposition isotopic changes to surface snow, and their assessment of relevance to the interpretation of annual ice core records (e.g. Renland) is not yet supported by the four separate 2-4 day field data expermients.

Specific Comments:

Note: Please see the line by line comments in the commented pdf. Their repetition here is duplicative, although I have pulled out some of key comments below:

Line 187: If this is the case, are you suggesting the results from the field experiments are only affected at the snow surface by sublimation as well, and the rest of the signal at depth is diffusion (below 0.5cm)?

We are suggesting that the laboratory experiments are only affected at the snow surface by sublimation, and the rest of the signal at depth is dominated by diffusion. In the case of the laboratory experiments, small fans maintain mixed air in the chamber, but do not blow air directly across or into the snow samples. However, in field conditions there are stronger wind speeds driving air through the upper several cm of the snowpack. This forced ventilation may cause increased sublimation at depth, or greater vapor exchange between the atmosphere and pore spaces between snow grains (Town et al., 2008). Therefore, the model more closely reflects conditions of the laboratory experiments, in which we are focusing on sublimation in a controlled environment.

We have revised L188 to clarify this: "...any isotopic changes below 0.5 cm can be attributed to diffusion (Ebner et al. 2017). In the laboratory experiments, because there is low airflow we can assume that there is no forced ventilation driving sublimation below the surface; however, this may not be the case for field conditions (Town et al., 2008)."

Line 284: It would be useful to run the snow isotope model with some of the field observations and input values and show how that compares to the model results from L5, which has a very high LHF that is continuous versus much lower mean LHF for FB or FS.

The snow isotope model is intended to simulate the effects of sublimation under dry air. In this case, it is known that any vapor in the system originated from the snow surface, and as such we can use a mass balance model to estimate the snow isotope value from the vapor isotopic composition.

In the case of the field experiments, there are two reasons why it is more difficult to model the snow composition from vapor measurements: 1) Over the course of the experiment, it is unknown what fraction of the vapor originated from the snow surface, and what fraction of the vapor was transported from elsewhere above the ice sheet or the ocean. Thus, our mass balance model may incorrectly estimate changes in the snow surface based on vapor that is not entirely sublimative flux from the snow surface. 2) Our mass balance model is developed based on a scenario in which sublimation is constantly occurring and there is no condensation. All field experimental periods have time periods with negative latent heat flux values, indicating condensation is occurring. Therefore, we cannot use the mass balance model for a full experimental period.

However, we do agree that in the context of this manuscript, it would be beneficial to demonstrate the usefulness of this model in a field setting. Therefore, we have run the model for the period from 6:00-18:00 on July 25 during F4. Throughout this period, the LHF is positive, indicating sublimation is occurring. In this case, we must assume that the measured vapor originates from the snow surface, which may not be true depending on the synoptic weather patterns. With the data we have, we are unable to speculate whether this is the case. However, as the vapor measurements are very close (~10cm) above the snow surface, the mixing state might be minor.

The following is implemented in the paper in L312 of the discussion:

"Figure X [shown here below] shows the mass balance model output for 6:00-18:00 on July 25. The starting snow isotopic composition is the same as the starting composition of the FB4 samples, and model input includes vapor measurements for d18O, d-excess, and humidity, as well as latent heat flux measurements and a linear estimate of temperature (as temperature data is not available for this period). We find that the surface d18O value increases by approximately 0.6 per mil, and the d-excess value decreases by approximately 1.3 per mil. The change propagates only 0.5 cm below the snow surface.

In comparison, the surface (0-0.5 cm) FB4 sample during this time period increases by approximately 1 per mil, and d-excess decreases by approximately 4 per mil (see Fig. A13). There is also a greater change in the sample from 0.5-1 cm, which increases by approximately 0.7 per mil in d18O, and decreases by approximately 2.5 per mil in d-excess in FB4 samples. The larger change observed in the field samples may be due to several factors, including the possibility that the measured vapor is a combination of sublimated vapor from the samples and vapor sourced from elsewhere. Additionally, wind pumping in the field setting may increase the rate of isotope change in the FB samples. As expected from the lab experiments, we see a similar trend towards enriched snow when running the model with actual field observations."

[Figure]

Line 288: Worth noting here that in the FS experiments depth propagation is only 1-2cm (max)

This is not a relevant comparison in L288, as the FS experiments take place over a shorter time period (2-2.5 days), and therefore it is expected that the depth propagation is lower than the laboratory experiments (4-6 days) and model (4 days).

Line 299: See comment from Line 284, and run the isotope box model with more realistic field conditions so in the discussion one can comment on the degree of sublimation impact in the field.

See response to previous comment.

Line 325: This one of the key questions, as the long term (weekly/monthly) result may not cause a significant deviation from the original snow-pack precipitation if the daytime sublimation and nighttime condensation of vapor balance each other out.  What is the NET change of the isotopic content over the entire month of July for the surface snow? Include in the Discussion.

With the data we currently have, we cannot differentiate between the long-term effects of precipitation and sublimation/deposition on the surface snow isotopic composition. Each of the experimental periods took place during a precipitation-free period, but there was precipitation between the experimental periods. In this study we do not have continuous snow surface data for the entire month of July, which would be required to determine the net isotope change due to precipitation vs. sublimation/condensation.

However, we do have continuous measurements of latent heat flux over the month of July, and there is an average LHF of 3.1 W/m2. This would indicate that daytime sublimation dominates over nighttime vapor deposition, and there would likely be a net increase in surface snow d18O over the full month. Since the reviewer asks specifically for speculations about the net influence of vapor exchange on the snow isotopic composition we have revised the paragraph from L358-370:

To assess the relevance of our results on longer timescales, we make use of the simple mass balance model and an observed mean latent heat flux in July of 3.1 W/m2, indicating a net removal of snow from the surface due to sublimation. By assuming  equilibrium fractionation during sublimation (Wahl et al2021) we can calculate the isotopic composition of the humidity flux and the associated removal of isotopologues. When considering reasonable values of a 5 cm layer of snow, a snow density of 300 kg/m3, an initial isotopic composition of -20‰ d18O and a surface temperature of -9 C for the month of July, the snow would be enriched by ~4‰ due to the net humidity flux which is substantial. For comparison, the seasonal amplitude (i.e.

summer peak to winter trough) at the Renland Ice Cap, for example, is about 8‰ in d18O (Hughes2020). We acknowledge that this is a highly simplified mass balance calculation without taking into account the vapor isotopic composition or precipitation inputs. However, since vapor exchange is a continuous process it will continuously affect the layer of snow that is in contact with the atmosphere, and will therefore imprint on the snow isotopic composition with a general net daily sublimation signal during months with a net sublimation flux.

Which months show a net sublimation flux is dependent on the geographical location and general climatology of the area. Especially in the context of paleoclimatological interpretation of ice cores, this cannot be assumed to be constant in time. If the vapor-snow exchange imprints on the seasonal snow isotopic composition as indicated in the result of the mass balance calculation, one would need to take changes in seasonality into account when making assumptions about vapor-exchange effects on paleo timescales, as has been previously demonstrated for precipitation seasonality (Werner et al. 2000).

On shorter timescales in our laboratory experiments we observe changes of up to 8 per mil d18O and 20 per mil d-excess over time periods of several days, and in FS field experiments we find an average change of 2.09 per mil d18O and 3.78 per mil d-excess on very short (sub-diurnal) timescales. This observation, in combination with our mass balance calculation of 4 per mil change in d18O over the month of July, suggests that under typical natural conditions, changes in the surface isotope value occurring on a short timescale may have an impact on the mean seasonally-recorded isotope signal. Previous studies have addressed the effect of seasonally-biased accumulation rate on diffusion and the recorded 18O isotope signal (Persson et al., 2011; Casado et al., 2020; Hughes et al., 2020) and the effect of physical modifications and snow redistribution of the snow surface on the accumulation intermittency (Zuhr et al., 2021), but the effect of sublimation driven changes in surface snow isotopic composition between precipitation events has not been quantified previously. Whether the magnitude of the mean isotope change due to sublimation and snow-vapor exchange outweighs the effects of snow redistribution, accumulation bias, and diffusion has yet to be determined.

Citation:

Werner, M., Mikolajewicz, U., Heimann, M., & Hoffmann, G. (2000). Borehole versus isotope temperatures on Greenland : Seasonality does matter. *Geophys. Res. Lett.*, *27*(5), 723–726.

Wahl, S., Steen-Larsen, H. C., Reuder, J., & Hörhold, M. (2021). Quantifying the Stable Water Isotopologue Exchange between Snow Surface and Lower Atmosphere by Direct Flux Measurements. *Journal of Geophysical Research: Atmospheres*, 1–24. https://doi.org/10.1029/2020jd034400

Line 350: The authors have not demonstrated that this is the case, as their field snow surface experiments on only on the order of 2-4 days, and they do not provide data from a snowpit at the end of the ~3 week sampling period that can support this statement.  It may or may not be superimposed on the precipitation signal, and therefore its an assumption that the "resulting snowpack record would be more indicative of atmospheric conditions..."

While a snowpit would be useful in future studies, we do know that over each individual 2-3 day experimental period there is no precipitation. Therefore, any changes observed in the surface snow over the experimental period is an atmospheric signal superimposed on the precipitation signal. Because we do not know the relative strength of the atmospheric signal vs. the precipitation signal, we have changed the wording of L352: "...resulting in a snowpack record indicative of multiple parameters including atmospheric conditions, water vapor isotopic

composition, condensation temperature (i.e. d18O), and precipitation source region conditions (i.e. d-excess)"

Figure A15. Include the RMSE or 2 sigma stdev for the FS1-4 data, and error bar on each graph, so that readers can view the uncertainty around the fit.

The RMSE has been added to Table 3, as requested in the later comment. We have also added the error bar for the linear regression +/- the standard deviation in Figure A15.

Technical Corrections:

Figure 8 caption. The color appears to be brown in the image. Change "FS surface snow 0-0.5 cm values are shown in dark orange" to brown

"Dark orange" has been changed to "brown".

Line 329: "In general, the box samples experience less decrease (should be increase) in d18O than associated FS samples due to minimized vapor deposition, and greater decrease in d-excess due to increased sublimation"

"Decrease" is correct here, as this sentence is referring specifically to periods of decreasing d18O when there is a negative LHF. This sentence has been revised to make this more clear: "In general, the box samples experience less decrease in d18O than associated FS samples due to minimized vapor deposition during periods of negative LHF, and greater total decrease in d-excess due to increased total sublimation across the entire experimental period."

Figure A13. Figure label says 2.5-4.5 cm (yellow), need to be consistent with Figure caption that states "2.5-4cm below the surface".

The figure caption has been corrected: "...at 2.5-4.5 cm below the surface."

Comments from annotated PDF:

L18: Use a qualifier: "minor" or "modest", as the impact is likely negligible over annual averaging in higher accumulation regions with more continuous snow fall.

L18 has been changed to: "...processes such as sublimation play a modest role in creating the climate signal…"

L20: Add ", particularly in regions of low accumulation."

L20 has been changed to: "...ice core climate record should be interpreted as such, particularly in regions of low accumulation."

Table 1: Note in the caption that L6-L8 were performed at a different lab.

The caption has been modified: "Overview of all experiments conducted. Eight Laboratory experiments were completed, with L1-L5 completed at the University of Colorado Boulder, and L6-L8 completed at the University of Copenhagen. Four field experiments (F1-F4) were completed at the East Greenland Ice Core Project field site. Field experiments included..."

L87: hopefully this can be moved to directly below Eq. 1. Almost seems like a Table caption where its located.

This placement is just a function of the preprint typesetting, and should be fixed in the final version.

L94: can you provide a comparison to polar air humidity like at the Greenland summit? In ppm

The mean polar air humidity for July 2019 at the Greenland summit is ~3400 ppmv (calculated from relative humidity provided at https://gml.noaa.gov/dv/data/index.php?site=sum&perpage=100&pageID=1&showall=1&frequency=Hourly%2BAverages¶meter_name=Meteorology).

Fig 1 caption: Can you provide arrows with labels on FS (blue photo) to make it very clear which FS samples are sampled at the respective depths? Thanks

Labels for depths of FS samples have been added to Figure 1.

L105: Cite: Schauer, Schoenemann, & Steig. Routine high-precision analysis of triple water-isotope ratios using cavity ring-down spectroscopy. RCMS, 2016

The citation for Schauer et al, 2016 has been added.

L120: Were the experimental setups and Picarro CRDS analyzers (water standards, protocols, etc.) identical between CU Boulder and UCopenhagen? This could potentially impact the results (depending on memory differences between instruments, etc.)

Water standards were different but had similar ranges (all standards are listed in Table A1), and the Picarro CRDS analyzers were different instruments but the same models. Calibration protocols would account for any differences between the individual instruments, and the experimental setups were otherwise identical.

L126: It would be helpful to have this in the same units as above for humidity (line 94,100ppm, or provide both specific and relative)

L94 has been changed to: "...resulting in humidity <100 ppm (i.e. <5% RH)."

L216: Clarify: Changes in the isotope signal are observed to propagate several cm into snow pack due to diffusion over X-period, driven by the induced …

L216 has been clarified: "Changes in the isotope signal are observed to propagate several cm into the snowpack due to diffusion over 4-6 days, driven by the induced…"

Table 2: It would be useful to provide the average of FS1-4 at the bottom of the table for both d18O and dxs, and can be used in the discussion, as its more representative to discuss the averages than the maximum value that occurred.

The mean of the isotope ranges for the experiments is: d18O = 2.09; d-excess = 3.78. This has been added to a bottom row of Table 2.

In the discussion L362-363 has been changed from "and in FS field experiments we find up to 3 per mil changes in d18O and 4 per mil changes in d-excess" to "and in FS field experiments we find an average change of 2.09 per mil d18O and 3.78 per mil d-excess"

Table 2: Based on Fig. 5, the range of FS2 0-0.5cm appears to only be ~2.4 permil, not 3. Can you double check?

The range of FS2 0-0.5 cm of 3 permil is correct. The minimum value is -25.12, and the maximum value is -22.12 permil.

L275: The sign of the covariation changes throughout the sampling period. For example, FS4, vapor d18O increases mid-day July 25, while the FS d18O decreases. If sublimation is occurring during a positive LHF period, then shouldn't the FS d18O snow surface become enriched (less negative d18O), as well as the vapor d18O? Can you discuss the difference in evolution of FS4 relative to the others?

Over the mid-day July 25 during F4, it is the case that the FS4 samples from 0.5-4 cm show decreasing d18O while the vapor d18O is increasing. This may be due to some variation in the subsurface layers due to precipitation or buried snow dunes from wind effects, which happens to be stronger in FS4 than in other experiments.

However, the surface 0-0.5 cm sample is increasing throughout most of the mid-day period. It is expected that the surface 0-0.5 cm sample would show the closest co-variation with vapor in comparison to the deeper samples. To make it more clear that the surface sample is of primary interest, the 0-0.5 cm dark brown line has been made bold in all subplots of Fig. 5.

Table 3: Recommended to include the RMSE for the 3hr snow sampling resolution for each experiment in a new column, which might show that FS4 had a very large amount of scatter around the fit.

We have added the RMSE for each experiment to the table, which is as follows:

FS1 = 1.62; FS2 = 1.67; FS3 = 1.60; FS4 = 1.39

Table 3: FS4: Although the P-value is significant, the R-value of 0.49, or R2 of 0.24 variance explained is a pretty low correlation. Despite the large diurnal variation in vapor d18O, the FS d18O does not seem to track it as expected, which is something to consider in the Discussion. See comment above about RMSE.

There are many factors which could potentially drive the relationship between the snow surface d18O and vapor d18O. As discussed in L334-345, it is difficult to isolate specific processes occurring in the field. Changes in the snow surface isotopes may be due to a combination of sublimation, vapor condensation, and other post depositional effects such as wind pumping and diffusion. For example, the deeper snow samples (0.5-4 cm) in FS4 exhibit a different pattern than in FS1-FS3, which may be due to buried wind dunes from precipitation events occurring in the time period between F3 and F4. Diffusion between the deeper layers and the surface 0-0.5 cm sample may alter the surface layer and reduce the strength of the relationship between the surface layer and the vapor isotopes (as discussed in L340-344).

The isotopic composition of atmospheric vapor is also driven by synoptic-scale variability, which may not be imprinted on the snow surface as rapidly as other processes such as sublimation.

With the limited data set currently available for four experiments, we still find a convincing argument in which three of the experiments show statistically significant relationships, even if one of these (FS4) has a slightly lower R value. This finding does not necessarily allow us to determine causality, but it does show that the atmospheric vapor and snow isotopic composition typically co-varies. Additional experiments in the future will help to strengthen this relationship and determine the factors driving changes in vapor vs. snow isotopic composition.

L348: Based on the depth profiles for extreme L1-L8 experiments, I would not state this as 2-3cm, but 1-2cm for field conditions

L348 "2-3 cm" has been changed to "1-2 cm".

L363: You have not shown that the short term uni-directional sublimation changes are preserved (or not overprinted) on a seasonal timescale, as demonstrated by the large change (decrease) in surface snow d18O values 2 weeks later in FS4.

This is correct, and this problem is described as potential future work in L371-376. To clarify this sentence, the strength of the wording in L363 has been decreased: "...changes in the surface isotope value occurring on a short timescale may have an impact on the mean seasonally-recorded isotope signal."

L393: add "(top 1-2 cm)"

L393 has been changed: "...significant changes in the top 1-2 cm of snow surface isotopes…"

L395: "substantial" is an over statement. use somewhat instead.

L395 "substantial" has been changed to "moderately"

L401: Provide a modifier for this statement, as it is still unclear whether the signal is preserved in the ice core record over a seasonal or yearly timescale.

L401 has been changed: "...but also may integrate the atmospheric conditions…"

---

## Author Comment (AC4)

Munch and Laepple short comment:

Dear Authors,

Let us first congratulate you on this interesting and elaborate study, combining lab experiments, modelling, and field studies, to advance our knowledge concerning one of the pressing questions in current ice core research which is the role of sublimation for the isotopic signal formation in polar snow and firn.

In the following we would like to comment on two aspects of your study: the use of the mass balance model and the interpretation of the field study data.

The lab experiment quite convincingly shows how under these controlled conditions sublimation goes along with isotopic fractionation in the surface snow and how this change propagates with depth due to diffusion. You compare your results to the output of a simple model which you drive with the observed isotopic composition of the vapour from your lab experiment together with a mass balance equation. Given this model setup, it is however not surprising that the model qualitatively reproduces the isotopic change in the surface snow, since you directly feed the model with the observational data that inherently includes already the fractionation effect from the sublimation. In our opinion it would be more convincing to use some fractionation model and the mass turnover to dynamically model both the evolution of the snow and the vapour isotopic composition.

One of the advantages of the laboratory experiment is the compliance with the closure assumption. We therefore argue that it is reasonable to use the measurements directly rather than some kind of fractionation model that might or might not be tunable to fit these observations. As you point out in your first sentence of the above paragraph, the observations of the enrichment of the snow under pure sublimation conditions serve the purpose of clearly demonstrating fractionation during sublimation and consequential enrichment in the snow. Albeit it would be easiest if traditional fractionation (i.e. Craig-Gordon) models could reproduce the experimental results, we find it not surprising that established evaporation models are unable to reproduce our observations, as evaporation from a water surface and sublimation from a porous snow surface show quite different characteristics.

In a further step of your study, you compare the hourly evolution of atmospheric vapour isotopic composition at EGRIP to isotopic data from surface snow samples and conclude that the atmospheric vapour isotopic composition drives the surface snow isotopes on these time scales. Here we are a bit puzzled about the assumed causal relationship, especially concerning the results from the lab experiment. In the lab experiment, sublimation "creates" vapour $\delta 18O$ from initially dry air. In nature, i.e. for the field data, you seem to assume it the other way around? Can you comment on this? If so, why do you not use the measured atmospheric $\delta 18O$ and sublimation rate to drive a model of surface snow $\delta 18O$ (+ diffusion) and compare this to the observations? This would be a real test of your hypothesis. If we understand your simple mass balance model (your Equation C2) correctly, to do so one could model the temporal evolution of $R_S$ (surface snow) given the measured time series of $R_E$ (atmospheric vapour) and latent heat flux LHF. The latter would provide the sublimation rate and thus the mass change per unit time of the surface snow.

We do find that the atmospheric vapor and surface snow isotopic composition co-vary, but we agree that we cannot determine a causal relationship at this time. As we state in L337 of the

discussion, "At this stage it is unclear whether LHF, vapor d18O, or another factor is influencing the snow surface, or whether the snow surface composition is driving vapor d18O."

However, our laboratory experiments do demonstrate that the snow surface influences the atmospheric isotopic composition during sublimation, and so we have now included a model run with field observation data as input for a very short time period in which sublimation is occurring. As the model is only designed to simulate the sublimation process, we are constrained to periods of continuous positive latent heat flux. Additionally, the field vapor isotopic composition that we use to drive the model is not only created solely from the snow but also subject to synoptic meteorological conditions. It would be interesting to use a C&G type model for the field experimental conditions but we argue that this would go beyond the intended scope of this paper. A larger data set will be necessary for this.

The following is implemented in the paper in L312 of the discussion:

"Figure X [shown here below] shows the mass balance model output for 6:00-18:00 on July 25. The starting snow isotopic composition is the same as the starting composition of the FB4 samples, and model input includes vapor measurements for d18O, d-excess, and humidity, as well as latent heat flux measurements and a linear estimate of temperature (as temperature data is not available for this period). We find that the surface d18O value increases by approximately 0.6 per mil, and the d-excess value decreases by approximately 1.3 per mil. The change propagates only 0.5 cm below the snow surface.

In comparison, the surface (0-0.5 cm) FB4 sample during this time period increases by approximately 1 per mil, and d-excess decreases by approximately 4 per mil (see Fig. A13). There is also a greater change in the sample from 0.5-1 cm, which increases by approximately 0.7 per mil in d18O, and decreases by approximately 2.5 per mil in d-excess in FB4 samples. The larger change observed in the field samples may be due to several factors, including the possibility that the measured vapor is a combination of sublimated vapor from the samples and vapor sourced from elsewhere. Additionally, wind pumping in the field setting may increase the rate of isotope change in the FB samples. As expected from the lab experiments, we see a similar trend towards enriched snow when running the model with actual field observations."

[Figure]

Concerning the field surface samples we would also welcome some more thoughts on the impact of isotopic spatial variability on your results. You already noted and partly accounted for the spatial variability by averaging across three samples. What are the horizontal variations in your data compared to the observed temporal change? One way to show this would be to add standard error estimates (from the 3 replicates) in Figure 5. Depending on the outcome it might

be useful to provide some suggestions on how to improve the field study setup in order to rule out confounding spatial with temporal changes.

The spatial variability of the field snow is one reason why the FB experiments with mixed snow were conducted as an intermediate version between laboratory and field.

We also try to keep spatial variability influences small by taking snow samples directly next to each other (see Fig. A1) and averaging across the three sample sites. In Fig. A12 you can see the spatial variability between the three sites and a clear offset in the mean is noticeable. Nevertheless, the individual sites can all experience a similar temporal evolution.

In future studies we would suggest to either take more samples from various locations to take into account the spatial variability OR exclude spatial variability concerns by digging a large pit (like 20cm x1m x 10m) and fill it with well-mixed snow. That way the starting isotopic composition would be the same for all samples (even in the vertical) but it would represent natural conditions closer than our field box experiments.

Finally, we would like to see some more elaboration in the discussion section of the impact of your results on longer time scales: How can you rule out that sublimation and subsequent deposition not just counteract and cancel each other on longer (seasonal or interannual) time scales? On the other hand, the effect could be relevant on palaeo time scales due to a stronger difference in relevant environmental conditions. In this view, the current comparison of your sub-diurnal change of 3 ‰ δ18O to a seasonal amplitude of ~8 ‰ δ18O (Renland) seems to be too simplistic. As an example, the temperature change in Berlin today between 4 am and 1 pm was 17 K which is the same amplitude as the annual cycle of temperature; still, it is unclear if the day-to-night changes have a significant impact on the summer-to-winter changes.

We have added a speculation of the snow-vapor exchange effect on monthly timescales based on the measured NET sublimation flux in July. This is a very simple mass balance calculation that finds a 5 cm snow layer is enriched by 4‰ in d18O during July. As you point out, whether there is a net sublimation or deposition flux is dependent on the local climatology. We mention this now also in the context of paleo timescales.

You are correct that no direct interpretation can be drawn from the comparison between the sub-diurnal change to seasonal change. However, it gives the reader a feeling for the magnitude of the observed changes that they would otherwise only have if they were very familiar with the topic. To elaborate on this, we have revised the paragraph from L358-370:

To assess the relevance of our results on longer timescales, we make use of the simple mass balance model and an observed mean latent heat flux in July of 3.1 W/m2, indicating a net removal of snow from the surface due to sublimation. By assuming equilibrium fractionation during sublimation (Wahl et al2021) we can calculate the isotopic composition of the humidity flux and the associated removal of isotopologues. When considering reasonable values of a 5 cm layer of snow, a snow density of 300 kg/m3, an initial isotopic composition of -20‰ d18O and a surface temperature of -9 C for the month of July, the snow would be enriched by ~4‰ due to the net humidity flux which is substantial. For comparison, the seasonal amplitude (i.e. summer peak to winter trough) at the Renland Ice Cap, for example, is about 8‰ in d18O (Hughes2020). We acknowledge that this is a highly simplified mass balance calculation without taking into account the vapor isotopic composition or precipitation inputs. However, since vapor exchange is a continuous process it will continuously affect the layer of snow that is in contact

with the atmosphere, and will therefore imprint on the snow isotopic composition with a general net daily sublimation signal during months with a net sublimation flux.

Which months show a net sublimation flux is dependent on the geographical location and general climatology of the area. Especially in the context of paleoclimatological interpretation of ice cores, this cannot be assumed to be constant in time. If the vapor-snow exchange imprints on the seasonal snow isotopic composition as indicated in the result of the mass balance calculation, one would need to take changes in seasonality into account when making assumptions about vapor-exchange effects on paleo timescales, as has been previously demonstrated for precipitation seasonality (Werner et al. 2000).

On shorter timescales in our laboratory experiments we observe changes of up to 8 per mil d18O and 20 per mil d-excess over time periods of several days, and in FS field experiments we find an average change of 2.09 per mil d18O and 3.78 per mil d-excess on very short (sub-diurnal) timescales. This observation, in combination with our mass balance calculation of 4 per mil change in d18O over the month of July, suggests that under typical natural conditions, changes in the surface isotope value occurring on a short timescale may have an impact on the mean seasonally-recorded isotope signal. Previous studies have addressed the effect of seasonally-biased accumulation rate on diffusion and the recorded d18O isotope signal (Persson et al., 2011; Casado et al., 2020; Hughes et al., 2020) and the effect of physical modifications and snow redistribution of the snow surface on the accumulation intermittency (Zuhr et al., 2021), but the effect of sublimation driven changes in surface snow isotopic composition between precipitation events has not been quantified previously. Whether the magnitude of the mean isotope change due to sublimation and snow-vapor exchange outweighs the effects of snow redistribution, accumulation bias, and diffusion has yet to be determined.

Citation:

Werner, M., Mikolajewicz, U., Heimann, M., & Hoffmann, G. (2000). Borehole versus isotope temperatures on Greenland : Seasonality does matter. *Geophys. Res. Lett.*, *27*(5), 723–726.

Wahl, S., Steen-Larsen, H. C., Reuder, J., & Hörhold, M. (2021). Quantifying the Stable Water Isotopologue Exchange between Snow Surface and Lower Atmosphere by Direct Flux Measurements. *Journal of Geophysical Research: Atmospheres*, 1–24. https://doi.org/10.1029/2020jd034400